# Ssu72 phosphatase is essential for thermogenic adaptation by regulating cytosolic translation

Eun-Ji Park [1,6], Hyun-Soo Kim [1,6] ✉, Do-Hyoung Lee[1], Su-Min Kim[1], Joon-Sup Yoon [1], Ji-Min Lee[2], Se Jin Im [3], Ho Lee [4], Min-Woo Lee [2] ✉ & Chang-Woo Lee [1,5] ✉

Brown adipose tissue (BAT) plays a pivotal role in maintaining body temperature and energy homeostasis. BAT dysfunction is associated with impaired metabolic health. Here, we show that Ssu72 phosphatase is essential for mRNA translation of genes required for thermogenesis in BAT. Ssu72 is found to be highly expressed in BAT among adipose tissue depots, and the expression level of Ssu72 is increased upon acute cold exposure. Mice lacking adipocyte *Ssu72* exhibit cold intolerance during acute cold exposure. Mechanistically, Ssu72 deficiency alters cytosolic mRNA translation program through hyperphosphorylation of eIF2α and reduces translation of mitochondrial oxidative phosphorylation (OXPHOS) subunits, resulting in mitochondrial dysfunction and defective thermogenesis in BAT. In addition, metabolic dysfunction in Ssu72-deficient BAT returns to almost normal after restoring Ssu72 expression. In summary, our findings demonstrate that cold-responsive Ssu72 phosphatase is involved in cytosolic translation of key thermogenic effectors via dephosphorylation of eIF2α in brown adipocytes, providing insights into metabolic benefits of Ssu72.

Adipose tissue plays an important role in responding to both changes in nutrient supply and ambient temperature to preserve whole-body energy homeostasis[1]. Brown adipose tissue (BAT) is a specialized thermogenic organ that produces heat to maintain body temperature in an adaptive process. Under physiological conditions, cold exposure increases the thermogenic activity of BAT, and further recruits beige or brite adipocytes with thermogenic capacity to white adipose tissue (WAT) depots[2,3]. Emerging evidence suggests that activation of these thermogenic adipose tissues can improve systemic metabolism[4–7]. Reduced activity or dysfunction of adipose tissues is related to impaired metabolic health[8,9]. Therefore, it is of great clinical importance to identify molecular mechanisms regulating the function of thermogenic adipose tissues.

In response to acute cold exposure, pre-existing brown adipocytes rather than newly recruited beige adipocytes are activated via the sympathetic nervous system to elevate body temperature in both humans and mice[10–13]. Brown adipocytes are rich in mitochondria, and they express higher levels of uncoupling protein 1 (UCP1) than white adipocytes[14]. The mitochondrial oxidative phosphorylation (OXPHOS) system, which is composed of electron transport chain (ETC), is a key functional unit in the mitochondria. It creates a proton gradient whose electromotive force drives ATP synthase[15]. When the proton

[1]Department of Molecular Cell Biology, Sungkyunkwan University School of Medicine, Suwon, Republic of Korea. [2]Department of Integrated Biomedical Science and Soonchunhyang Institute of Medi-bio Science (SIMS), Soonchunhyang University, Cheonan, Republic of Korea. [3]Department of Immunity, Sungkyunkwan University School of Medicine, Suwon, Republic of Korea. [4]Graduate School of Cancer Science and Policy, Research Institute, National Cancer Center, Goyang, Republic of Korea. [5]Department of Health Sciences and Technology, SAIHST, Sungkyunkwan University, Seoul, Republic of Korea. [6]These authors contributed equally: Eun-Ji Park, Hyun-Soo Kim. ✉e-mail: jazz7780@skku.edu; mwlee12@sch.ac.kr; cwlee1234@skku.edu

conductance of UCP1 is increased in the inner mitochondrial membrane, UCP1 uncouples mitochondrial respiration from ATP synthesis and dissipates energy as heat[12]. Although UCP1-independent thermogenic pathway has been recently identified in thermogenic adipocytes[16], uncoupling of ATP synthesis in mitochondria remains a major mechanism of BAT thermogenesis[14,17]. The thermogenic function of BAT is critical for maintaining homeostatic body temperature in mammals[17]; however, it remains unclear how multiple thermogenic mechanisms are coordinately regulated in BAT.

The dramatic metabolic adaptation of BAT involves its action in responding to metabolic demand by coordinating cellular organelles (such as mitochondria and endoplasmic reticulum (ER)) and by controlling abundance of lipids and proteins[18]. Disruptions in ER homeostasis can trigger unfolded protein response of ER (UPR$^{ER}$) and regulate transcription and translation to alleviate the unfolded protein load and expand the folding capacity[19,20]. Because fine-tuned protein synthesis is crucial for sustaining cell viability and function, dysregulation of UPR$^{ER}$ has been implicated in the pathogenesis of many diseases including metabolic diseases[19,21]. UPR$^{ER}$ consists of three branches controlled by inositol requiring protein 1α (IRE1α), PKR-like ER-regulated kinase (PERK), and activating transcription factor 6 (ATF6). PERK phosphorylates the α subunit of eukaryotic initiation factor 2 (eIF2α) at serine 51 (Ser51)[19]. Importantly, the phosphorylation of eIF2α represses the initiation phase of translation, mediating global decrease of cytosolic translation[22]. Due to the evolution of dynamically modulating ER stress signaling with different tissue distributions and activation profiles, studies on eIF2α phosphatases have been expanded[22,23]. Dephosphorylation eIF2α by inducible expression of eIF2α phosphatase GADD34 in the liver can improve insulin sensitivity and reduce hepatosteatosis in mice fed a high-fat diet (HFD)[24], and GADD34-deficient mice become obese with fatty liver by aging and HFD[25]. GADD34-deficient mice also exhibit resistance to diet-induced obesity with reduced food intake[26], consistent with an effect of hypothalamic eIF2α phosphorylation in regulating feeding behaviors[27]. In addition, it has been reported that protein-tyrosine phosphatase 1B (PTP1B) is involved in ER stress signaling[28] and PTP1B deficiency attenuates protein synthesis in brown adipocytes by increasing PERK phosphorylation[29]. However, little is known about how translational regulation through eIF2α phosphorylation directly affects metabolic adaptation of BAT.

Ssu72 is a dual-specific protein phosphatase that is expressed in a tissue-specific manner[30]. Recent studies have demonstrated that Ssu72 dephosphorylates both Ser 5 and Ser 7 in the carboxyl-terminal domain (CTD) of RNA polymerase II (RNAPII), thus playing an important role in controlling CTD function during the transcription cycle[31–33]. However, Ssu72 also exerts RNAP II-independent phosphatase activity in a tissue-specific manner involving epithelial cells, immune cells, and hepatocytes, thereby affecting physiological function and pathogenesis[30,34–36]. We have previously found that the activity of Ssu72 profoundly affects the maintenance of hepatic chromosome integrity, and it can be used to monitor the development of liver diseases, including non-alcoholic fatty liver disease (NAFLD), fibrosis, and steatohepatitis-associated hepatocellular carcinoma (HCC)[30,37]. Although some evidence for the role of Ssu72 in the pathogenesis of metabolic diseases has been found, its functions in major metabolic tissues such as adipose tissue are still largely unknown.

We initially found that Ssu72 was elevated in BAT and inguinal WAT (iWAT), and it was expressed much higher in BAT. Importantly, the level of Ssu72 was upregulated upon acute cold exposure, implying that Ssu72 might play a role in the thermogenic function of adipose tissues. We found that adipocyte-specific Ssu72 ablation resulted in mitochondrial dysfunction and cold intolerance in response to cold exposure. Furthermore, Ssu72 affected protein synthesis required for thermogenesis by directly regulating the function of eIF2α through hypophosphorylation. Taken together, these results revealed that

Ssu72 phosphatase plays an important role in cytosolic translation affecting mitochondrial oxidative phosphorylation and thermogenesis in brown adipocytes.

## Results

### Ssu72 is highly expressed in BAT and induced by cold exposure

To investigate the clinical significance of Ssu72 expression in adipose tissue, we initially compared *Ssu72* levels in isolated adipocytes between lean and non-diabetic obese individuals from Gene Expression Omnibus (GEO) public genomics database (GSE2508). Intriguingly, we found significantly decreased expression of *Ssu72* in the isolated adipocytes from obese individuals (Supplementary Fig. 1a, b). To explore the role of Ssu72 phosphatase in adipose tissue metabolism, we thus assessed the expression and distribution of selected representative phosphatases (serine/threonine phosphatase and dual specificity phosphatase) in various tissues including different fat depots. Surprisingly, Ssu72 phosphatase was highly enriched in brown adipose tissue (BAT) depot relative to white adipose tissue (WAT) depots, including inguinal WAT (iWAT) and epididymal WAT (eWAT) (Fig. 1a, b). Since the major function of BAT is thermogenesis, we examined endogenous difference in adipose Ssu72 expression in response to cold exposure. Notably, *Ssu72* mRNA level was increased significantly in BAT within 4 h after mild cold (15 °C) exposure, in parallel to relative *Ucp1* mRNA level (Fig. 1c and Supplementary Fig. 2a). In addition, the level of *Ssu72* in iWAT was increased significantly at 12 h after mild cold (15 °C) exposure (Supplementary Fig. 2b). We further assessed whether Ssu72 protein expression was increased during severe cold exposure. Thermogenic adipose tissues, iWAT and BAT, were taken from wild-type (C57BL/6J) mice housed at room temperature (RT, 23 °C) or exposed to acute severe cold (4 °C) for 4 h. Of note, the expression of Ssu72 protein in BAT was significantly increased after acute cold exposure (Fig. 1d). This indicates that Ssu72 might play an essential role in BAT thermogenesis. Moreover, Ssu72 protein levels in BATs were greater than those in iWATs at 4 h after cold exposure (Supplementary Fig. 2c), similar to our findings after housing mice at RT. Given the increased *Ssu72* mRNA level in iWAT at 12 h of mild cold exposure, we further examined the induction of Ssu72 protein in iWAT over a longer period of acute cold exposure. The expression of Ssu72 protein was increased in iWAT after cold exposure for 8–12 h (Supplementary Fig. 2d), implying that Ssu72 might also have a thermogenic function in iWAT during chronic cold exposure.

### Ssu72 is required for the maintenance of thermogenic adipose tissue

To identify the repercussion of increased Ssu72 level in thermogenic adipose tissues, we determined whether Ssu72 was involved in the maintenance and function of adipose tissues. To this end, we generated *Ssu72^{flox/flox}*; *Adiponectin-Cre* (hereafter Ssu72 aKO) mouse with Ssu72 deleted specifically in adipocytes by intercrossing *Ssu72^{flox/flox}* (hereafter Ssu72 WT) mice with *Adiponectin-Cre* mice (Fig. 1e and Supplementary Fig. 3a–c). Resulting Ssu72 aKO mice were born in normal Mendelian ratio. They were apparently indistinguishable from their littermates of Ssu72 WT mice. Body weight gain was also similar between genotypes from 6 to 20 weeks of age (Supplementary Fig. 3d). Surprisingly, BAT from Ssu72 aKO mice appeared pale compared to BAT from Ssu72 WT mice, even when mice were fed a normal chow diet and housed at RT (23 °C) (Fig. 1f). Total BAT weights were similar between genotypes (Supplementary Fig. 3e). In addition, Ssu72-deficient BAT exhibited increased conversion of multilocular into unilocular adipocytes, which was highly correlated with whitening of brown fat (Fig. 1g). Both iWAT and eWAT between Ssu72 WT and aKO mice at 6 and 22 weeks of age were very similar in size (Supplementary Fig. 3f). However, hematoxylin-eosin (H&E) staining and size analyses showed that adipocyte size and lipid droplet of iWAT from Ssu72 aKO

 

mice were increased compared to those from Ssu72 WT mice (Supplementary Fig. 3g). There was no significant difference in adipocyte size or lipid droplet of eWAT between Ssu72 WT and aKO mice (Supplementary Fig. 3h). In addition, tissue staining analysis using embryos at 15.5 and 18.5 days (E15.5 and E18.5, respectively) showed no significant difference in BAT morphology or Ucp1 expression in embryos between genotypes (Supplementary Fig. 4a), indicating that Ssu72 might not be involved in BAT development (Supplementary Fig. 4a). Notably, there were no significant changes in expression levels of general adipogenic genes such as *Fabp4* and *Pparg1* in BATs, between WT and aKO mice housed at RT (Supplementary Fig. 4b). Taken together, these results imply that

adipose Ssu72 is involved in the maintenance and thermogenic function of BAT.

## Ssu72 deficiency in BAT decreases thermogenic activation and fatty acid oxidation during acute cold exposure

Given the abnormal histological appearance of Ssu72-deficient BAT, we assume that adipocyte Ssu72 may be required for response to acute cold exposure. To examine thermogenic activity in Ssu72-deficient BAT, 7-week-old Ssu72 WT and aKO mice were challenged with acute cold exposure (4 °C). No significant change was seen in resting body temperature between the genotypes (Fig. 2a, time = 0). Importantly, in response to cold exposure, Ssu72 WT mice were able to maintain their

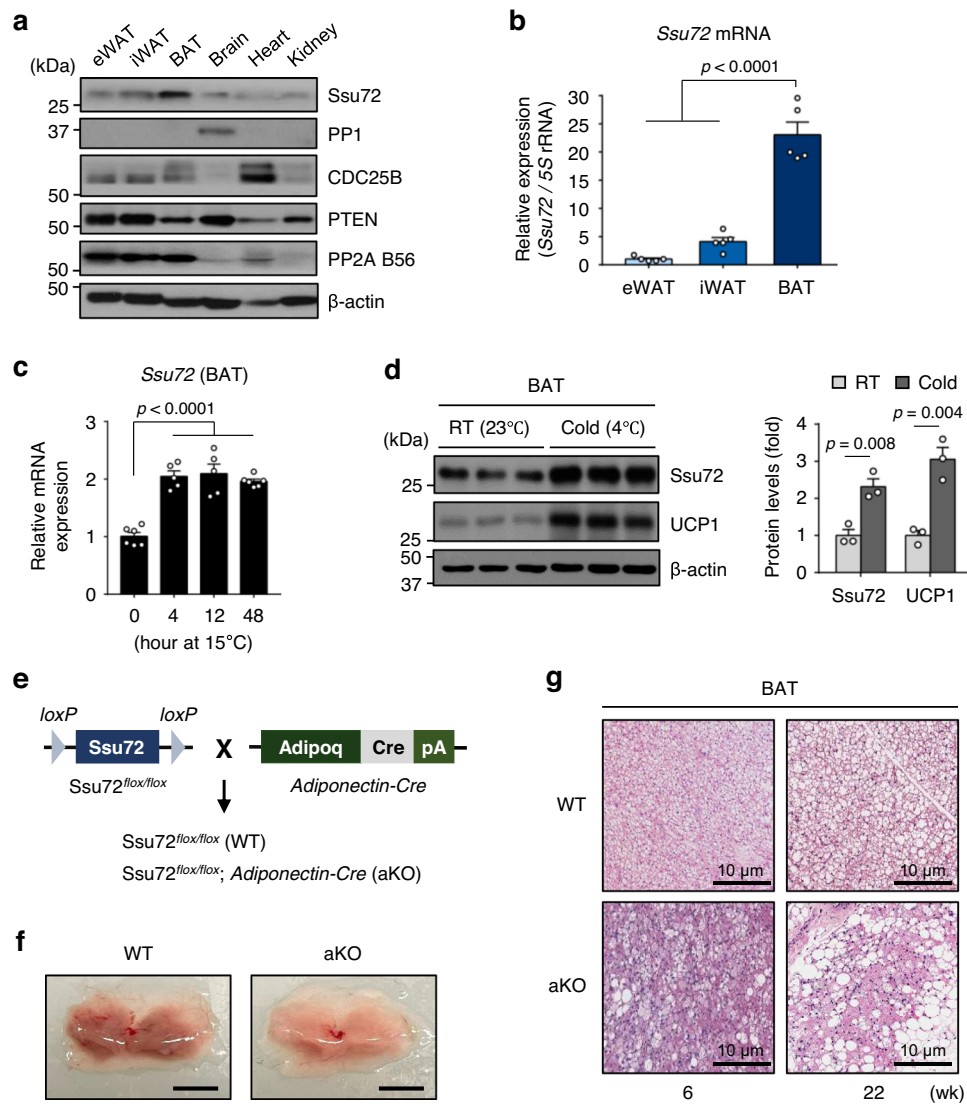

**Fig. 1 | Ssu72 phosphatase is potentially involved in physiological regulation of thermogenic adipose tissues, especially BAT. a** Various mouse tissues including epididymal white adipose tissue (eWAT), inguinal white adipose tissue (iWAT), and brown adipose tissue (BAT) were extracted from 7–8 weeks old C57BL/6 (WT) male mice, and lysates from tissues were immunoblotted with antibodies against several representative phosphatases, which were classified into four phosphatase groups (*n* = 5, biologically independent samples). **b** Relative mRNA expression levels of *Ssu72* in eWAT, iWAT, and BAT from 9-week-old WT mice (*n* = 5 mice per group). Data are presented as mean ± SEM. Statistical comparisons were made using one-way ANOVA. **c** Relative mRNA expression levels of *Ssu72* in BAT of 7 to 8-week-old WT mice after mild cold exposure (15 °C) (*n* = 5 mice per group). Data are presented as mean ± SEM. Statistical comparisons were made using one-way ANOVA. **d** Western blots of lysates from BAT of WT mice housed at RT (23 °C) or exposed to

acute cold (4 °C) for 4 h (left) (*n* = 3 mice per group), with quantification of Ssu72 and UCP1 protein levels (right). Graph shows quantification of Ssu72 protein levels normalized to β-actin protein level using ImageJ. Data are presented as mean ± SEM. Statistical comparisons were made using two-tailed Student's *t* test. **e** Animated images of Ssu72*flox/flox* (Ssu72 WT), *Adiponectin-Cre*, and *Ssu72flox/flox*; *Adiponectin-Cre* mice (Ssu72 aKO). **f** Representative images of BAT from 11-week-old Ssu72 WT and aKO male mice (scale bar, 5 mm; *n* = 10 per group, biologically independent samples from 8–11 weeks old mice). **g** Hematoxylin and eosin (H&E) staining of representative sections of BAT from 6- and 22-weeks-old Ssu72 WT and aKO male mice (*n* = 5 per group, biologically independent samples). For (**f**) and (**g**), both male and female mice were used, and no sex-specific phenotype was observed. Source data are provided as a Source Data file.

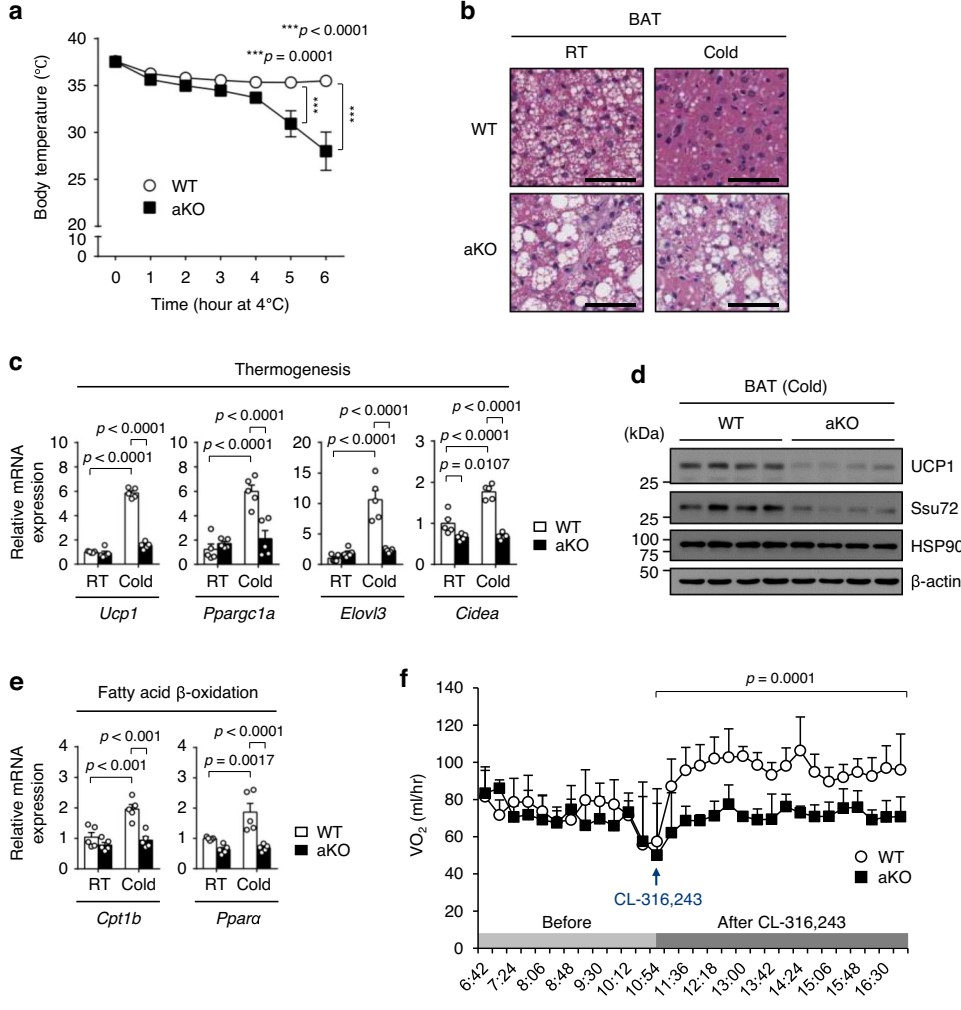

**Fig. 2 | Adipose Ssu72 deficiency results in cold intolerance and thermogenic defect of BAT. a** Rectal core body temperatures of 7-week-old Ssu72 WT and aKO male mice under acute cold conditions (4 °C) at indicated time points ($n = 8$ mice per group). Data are presented as mean ± SEM. Statistical comparisons were made using two-way ANOVA. **b** H&E staining of BAT from 7-week-old Ssu72 WT and aKO male mice housed at RT or exposed to acute cold (4 °C) for 4 h (scale bar, 100 μm; $n = 5$ per group, biologically independent samples). **c** Relative mRNA expression levels of thermogenic genes (*Ucp1*, *Ppargc1α*, *Elovl3*, and *Cidea*) in BAT from Ssu72 WT and aKO mice exposed to RT or acute cold (4 °C) ($n = 5$ mice per group). Data are presented as mean ± SEM. Statistical comparisons were made using two-way ANOVA. **d** Western blots of Ucp1 and Ssu72 in BAT from Ssu72 WT and aKO mice exposed to acute cold (4 °C) ($n = 4$ mice per group). **e** Relative mRNA expression levels of fatty acid β-oxidation-related genes (*Cpt1b* and *Ppara*) in BATs of Ssu72 WT and aKO mice exposed to RT or acute cold ($n = 5$ mice per group). Data are presented as mean ± SEM. Statistical comparisons were made using two-way ANOVA. **f** Whole-body oxygen consumption ($VO_2$) of 7-week-old Ssu72 WT and aKO male mice treated with 10 mg/kg CL-316,243 ($n = 5$ mice per group). Data are presented as mean ± SD. Statistical comparisons of mean $VO_2$ between WT and aKO mice under basal (before) and CL-316,243 stimulation conditions (after) were made using two-way ANOVA. Source data are provided as a Source Data file.

body temperature at around 36.5 °C after an initial drop of approximately 1.5 °C; however, Ssu72 aKO mice showed significant sensitivity to cold temperature. In Ssu72 aKO mice, the rectal temperature was significantly dropped within 4 h, dropped to below 30 °C in less than 6 h after cold exposure, and fatal hypothermia was found after 6 h of cold exposure (Fig. 2a). The BAT from Ssu72 aKO mice showed multiple interspersed cells, each containing a single lipid droplet in both RT and cold conditions (Fig. 2b), indicating that Ssu72 aKO mice exhibited defects in acquiring the active brown adipocyte phenotype.

The thermogenic function of BAT is known to be associated with the expression of several BAT thermogenic genes[38]. We thus compared mRNA expression levels of thermogenic genes (*Ucp1*, *Ppargc1α*, *Elovl3*, and *Cidea*) between BATs from Ssu72 WT and aKO mice maintained at RT or exposed to cold (4 °C) for 4 h. Under acute cold exposure, expression levels of genes related to BAT thermogenesis were significantly decreased in BAT of Ssu72 aKO mice than Ssu72 WT mice (Fig. 2c). Consistent with lower mRNA expression of thermogenic

genes, we observed a decreased expression of UCP1 protein in Ssu72 aKO mice compared to Ssu72 WT mice (Fig. 2d). Notably, cold-induced stimulation of β3-adrenergic receptor (β3-AR) leads to activation of mitochondrial fatty acid oxidation (β-oxidation) and increases mitochondrial biogenesis in BAT[39,40]. To investigate whether Ssu72 was linked to the signaling pathway of fatty acid oxidation, we measured expression levels of fatty acid β-oxidation-related genes in both RT and acute cold conditions. While acute cold exposure induced increases of expression levels of genes involved in fatty acid β-oxidation (*Cpt1b* and *Ppara*) in Ssu72 WT BAT, cold-induced expression of these genes was significantly attenuated in Ssu72 aKO BAT (Fig. 2e).

To determine the physiological contributions of Ssu72 in systemic whole-body energy expenditure, we housed Ssu72 WT and aKO mice individually in metabolic cages and measured metabolic parameters continuously during ad libitum feeding condition. Consistent with a requirement for thermogenesis during cold stimulation, an injection of CL-316,243 (a highly selective β3-adrenergic receptor agonist)

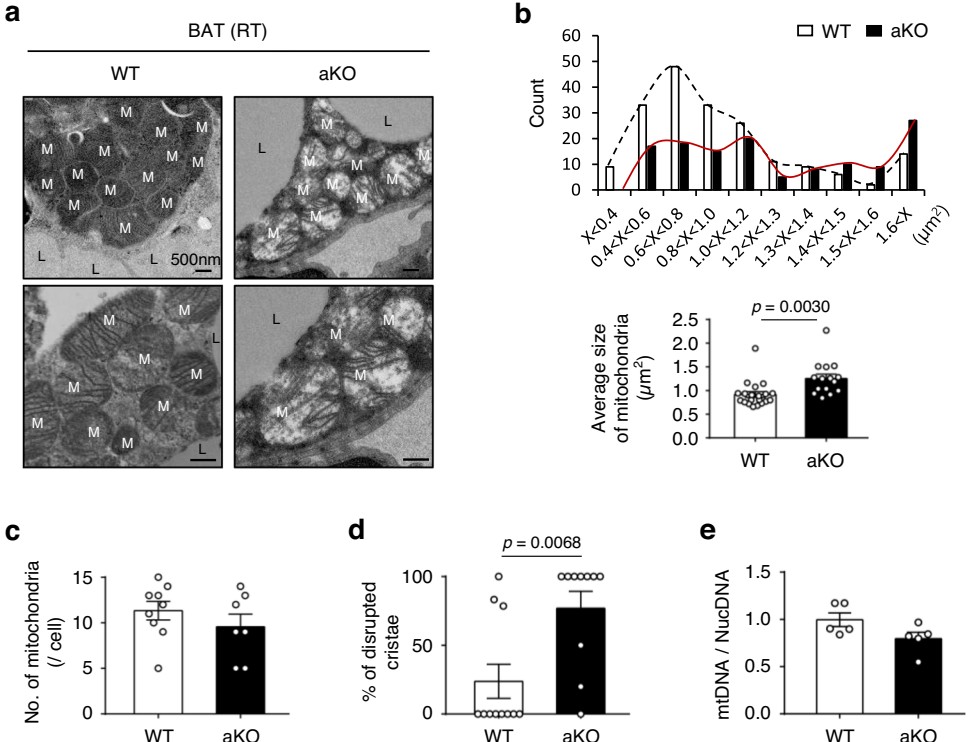

**Fig. 3 | Ablation of Ssu72 disrupts mitochondrial structure in BAT. a** TEM analysis of BAT from 8-week-old WT and aKO male mice housed at RT (scale bar, 500 nm). M and L refer to mitochondria and lipid droplets, respectively (WT, $n = 5$; aKO, $n = 4$). Both male and female mice were used for TEM analysis, and no sex-specific phenotype was observed. **b** Quantitative analyses of mitochondrial size distribution (upper panels) and average size of mitochondria in BAT from Ssu72 WT ($n = 20$) and Ssu72 aKO ($n = 15$) mice (bottom panels). Data are presented as mean ± SEM. Statistical comparisons were made using two-tailed unpaired Student's $t$ test. **c** Quantitative morphometric measurements of the number of mitochondria per cell (WT, $n = 9$; aKO, $n = 7$). Data are presented as mean ± SEM.

Statistical comparisons were made using two-tailed unpaired Student's $t$ test ($p = 0.3092$). **d** The percentage of disrupted cristae based on TEM images of BAT over 10 electron micrograph sections (WT, $n = 11$; aKO, $n = 10$). Data are presented as mean ± SEM. Statistical comparisons were made using two-tailed unpaired Student's $t$ test. **e** Quantitative PCR analysis of mitochondrial DNA (mtDNA) copy number normalized to nuclear DNA (NucDNA) from BAT of Ssu72 WT and aKO mice ($n = 5$ mice per group). Data are presented as mean ± SEM. Statistical comparisons were made using two-tailed unpaired Student's $t$ test ($p = 0.078$). Source data are provided as a Source Data file.

rapidly increased oxygen consumption in Ssu72 WT mice. However, it was completely ineffective in altering oxygen consumption in Ssu72 aKO mice (Fig. 2f). Additionally, there were no significant changes of respiratory exchange ratio (RER) and locomotor activity in Ssu72 aKO mice compared to controls (Supplementary Fig. 5a, b). Unlike ACC2 lacking mice model where changes of fatty oxidation drive alterations in feeding behavior[41], Ssu72 aKO mice had no change in food intake (Supplementary Fig. 5c), implying that reduction of energy expenditure in Ssu72 aKO mice might reflect physiological dysfunctions of adipose fatty acid oxidation, but not a compensatory requirement of energy expenditure from skeletal muscle. These results indicate that adipose Ssu72 contributes to fatty acid oxidation and whole-body energy expenditure to maintain core body temperature during cold stimulation.

### Loss of Ssu72 alters mitochondrial morphology in BAT

Defects in the thermogenic function of BAT are often closely linked to functional changes of mitochondria[42]. As noted above, a white-like phenotype in Ssu72-deficient BAT and a rapid decrease in body temperature of Ssu72 aKO mice during acute cold exposure (4 °C) suggest that thermogenic activation of Ssu72-deficient BAT is already disrupted even at ambient temperature. In fact, housing mice at RT (23 °C) is considered a mild cold exposure because this temperature is below thermoneutrality (30 °C), thus affecting the basal metabolic rate of mice[43]. Therefore, we investigated the probability of mitochondrial dysfunction in BAT of Ssu72 aKO mice housed at RT (23 °C). Interestingly,

transmission electron microscopy (TEM) analysis revealed aberrant mitochondrial structure in the BAT of Ssu72 aKO mice housed at RT (Fig. 3a). BAT of Ssu72 aKO mice showed enlarged mitochondria and disorganized cristae structures in comparison with Ssu72 WT mice, but there was no difference in the number of mitochondria between Ssu72 WT and aKO mice (Fig. 3b–d). Thus, Ssu72 appears to be related to maintaining mitochondrial cristae structure in BAT at ambient temperature. Previous studies have reported that disruption of mitochondrial dynamics affects the maintenance of mitochondrial DNA (mtDNA) copy number or integrity[44]. Despite aberrant changes in cristae structure, mtDNA copy number showed no significant difference between genotypes at room temperature (Fig. 3e), suggesting that disorganization of cristae caused by Ssu72 depletion did not directly affect mtDNA copy number.

### Ssu72 is involved in regulation of mitochondrial homeostasis, fatty acid metabolism, and UPR^ER signaling pathways

To understand the underlying mechanism of the defect in BAT conferred by Ssu72 depletion, we employed RNA-sequencing (RNA-seq) and performed signaling enrichment analysis. Total RNA of BAT from Ssu72 WT and Ssu72 aKO mice housed at RT were prepared for analysis. The FPKM value was used to quantify mRNA expression. We analyzed profiles of differentially expressed genes (DEGs) between genotypes. The analysis of DEGs (≥2-fold change (FC)) identified a total of 1919 DEGs in the Ssu72 aKO group compared to the Ssu72 WT group, including 1,006 upregulated genes and 913 downregulated

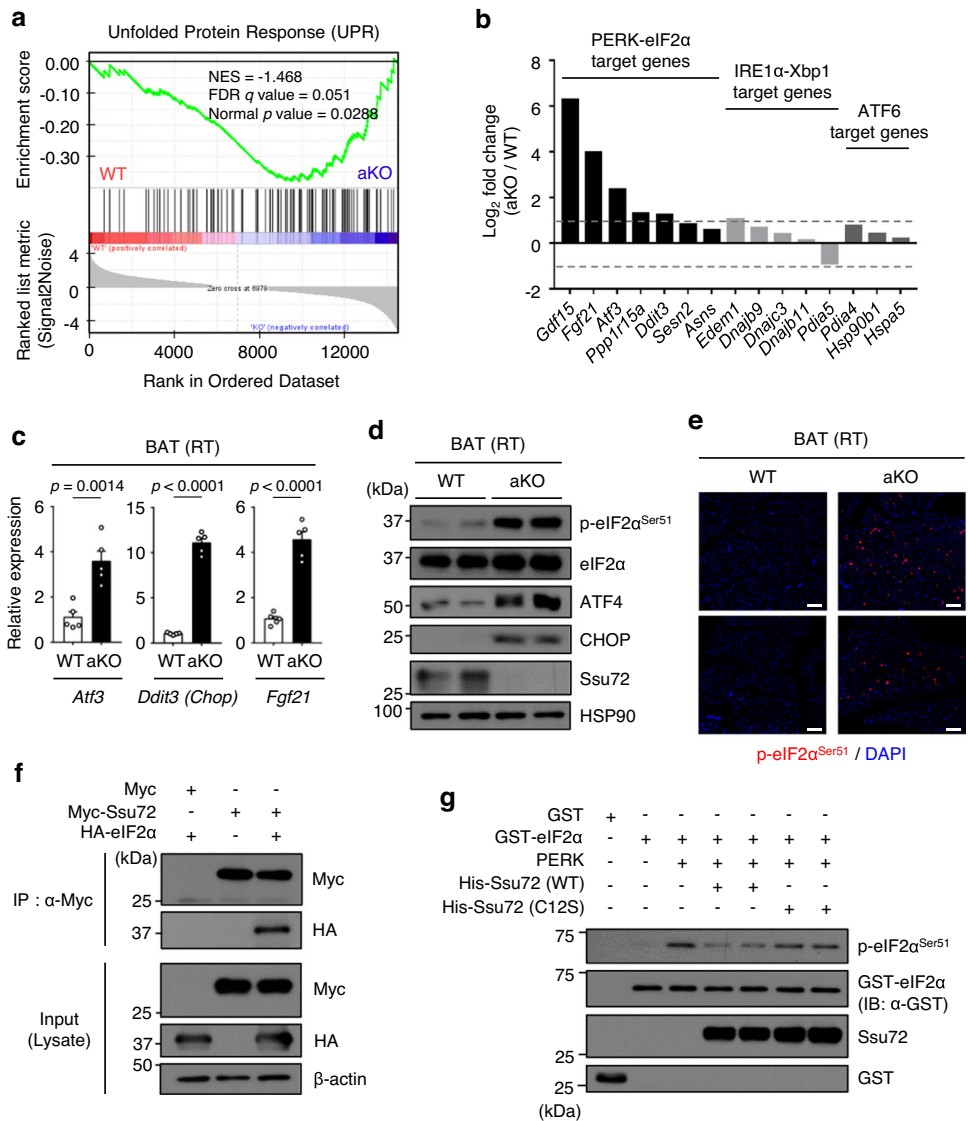

**Fig. 4 | Ssu72 phosphatase controls eIF2α signaling but not IRE1α and ATF6 signaling. a** Gene set enrichment analysis (GSEA) of hallmark unfolded protein response (UPR) gene sets in 8-week-old BAT of WT and aKO male mice ($n = 3$ mice per group). Green curve shows the enrichment score, which reflects the degree to which each gene is enriched (black vertical lines). A GSEA algorithm was used to evaluate the statistical significance of gene expression and calculate normalized enrichment scores (NES). Nominal $p$ value was calculated as two-sided $t$-test. FDR, false discovery rate. **b** Relative fold changes expressed as $\log_2$ of normalized average count (RNA-seq) of each gene in aKO BAT samples divided by that in WT BAT ($n = 3$ mice per group). Gray dashed lines indicate $\log_2$ fold changes of 1 or −1. **c** Relative mRNA expression levels of ATF4 target genes (*Atf3, Ddit3, Fgf21*) ($n = 5$ mice per group). Data are presented as mean ± SEM. Statistical comparisons were made using two-tailed unpaired Student's $t$ test. **d** Western blots in BAT from 10-week-old Ssu72 WT and aKO male mice housed at RT ($n = 2$ mice per group).

**e** Immunofluorescent (IF) staining of P-eIF2α$^{Ser51}$ (red) and DAPI (blue) in BAT from 11-week-old WT and aKO female mice (Scale bar, 100 μm) ($n = 5$ per group, biologically independent samples). **f** Co-immunoprecipitation (Co-IP) assays from 293T cells transfected with expression vectors for Myc, Myc-tagged Ssu72 (Myc-Ssu72) and HA-tagged eIF2α (HA-eIF2α). Lysates were immunoprecipitated with anti-Myc antibody. Immunoprecipitated Ssu72 was tested for interaction with eIF2α ($n = 4$ independent biological replicates). Panels below present western blots of input proteins or β-actin loading control and quantitative scans of amounts of each protein detected in western analysis of input protein panels. **g** In vitro phosphatase assay was performed with full-length glutathione S-transferase (GST)-fused phosphorylated eIF2α and recombinant His-Ssu72. Total eIF2α and P-eIF2α$^{Ser51}$ were analyzed by immunoblotting ($n = 3$ independent biological replicates). IP immunoprecipitation, IB Immunoblotting. Source data are provided as a Source Data file.

genes. Based on false discovery rate (FDR)-adjusted 0.05 *p*-value threshold, 819 genes were downregulated and 889 genes were upregulated among 1,708 DEGs ($p < 0.05$, FDR < 0.1) in Ssu72-deficient BAT than in WT BAT (Supplementary Fig. 6a). Enrichment analysis of gene ontology (GO) cellular component and biological process of upregulated and downregulated genes in BAT of Ssu72 aKO mice relative to BAT of Ssu72 WT mice was then performed. As expected, GO cellular component analysis revealed that downregulated genes of Ssu72 aKO BAT were predominantly involved in mitochondria and respiratory ETC. Downregulated genes enriched in

GO biological process were stratified into categories of oxidation-reduction process, mitochondrial translation, oxidative phosphorylation, and particularly the regulation of eIF2α phosphorylation (Supplementary Fig. 6b, upper panels). We next assessed genes upregulated in Ssu72-deficient BAT compared to those in WT BAT. The GO biological process analysis showed that fatty acid biosynthesis genes were upregulated in Ssu72 aKO BAT. Unexpectedly, ER unfolded protein response (UPR$^{ER}$) genes were also upregulated in Ssu72 aKO BAT compared to those in Ssu72 WT BAT, and cellular component analysis showed that the ER membrane and integral

component of ER were upregulated in Ssu72-deficient BAT (Supplementary Fig. 6b, bottom panels).

## Ssu72 dephosphorylates eIF2α in BAT

We next performed gene set enrichment analysis (GSEA) using RNA-seq results to examine whether Ssu72 was involved in UPR[ER] signaling pathways. GSEA showed significant enrichment of UPR[ER] genes within ranked gene expression of Ssu72 aKO BAT compared with Ssu72 WT BAT (Fig. 4a). To further examine the activation of three ER stress sensors of UPR, PERK, IRE1, and ATF6, we compared levels of phosphorylated PERK and IRE1 (P-PERK[Thr980] and P-IRE1[Ser724]) and cleaved ATF6 in BAT from Ssu72 WT and aKO mice. Although UPR-related genes were upregulated in Ssu72 aKO BAT compared to those in Ssu72 WT BAT, activation of three stress sensors of UPR[ER] showed no significant differences (Supplementary Fig. 7a), suggesting that there might be activation of downstream molecules involved in UPR signaling pathway rather than upstream ER stress sensors. To explore changes in the expression of specific target genes in the UPR[ER] signaling pathways by Ssu72 deficiency in BAT, we calculated the relative log$_2$-fold normalized average count (RNA-seq) values of selected genes induced by activation of three branches (PERK-eIF2α, IRE1α-Xbp1 and ATF6 pathway) of UPR[ER]. An interesting observation was that the majority of PERK-eIF2α target genes (*Gdf15, Fgf21, Atf3, Ppp1r15a,* and *Ddit3*) were significantly upregulated (log$_2$-fold change ≥1, $p < 0.05$) in Ssu72 aKO BAT than in WT BAT (Fig. 4b). Consistent with RNA-seq results, we also found that mRNA levels of eIF2α target genes (*Atf3, Ddit3,* and *Fgf21*) were clearly upregulated in Ssu72 aKO BAT (Fig. 4c).

To assess whether Ssu72 could regulate eIF2α phosphorylation in BAT, we conducted immunoblotting analysis in BAT of Ssu72 WT and aKO mice housed at RT. Surprisingly, phosphorylation of eIF2α at serine 51 was sharply increased in the BAT of Ssu72 aKO mice compared to that in the BAT of Ssu72 WT mice (Fig. 4d). In addition, protein expression levels of both ATF4 and CHOP induced by eIF2α phosphorylation were also increased in Ssu72 aKO BAT (Fig. 4d). To determine whether the increase in eIF2α phosphorylation shown above occurred in the whole BAT tissue, immunofluorescence (IF) staining was performed using BAT frozen sections of Ssu72 WT and aKO mice. We observed a substantial increase in eIF2α phosphorylation at serine 51 in whole-tissue section of Ssu72 aKO BAT (Fig. 4e), indicating that Ssu72 was essential for eIF2α signaling pathway. In mice exposed to acute cold (4 °C), we also observed an increase of eIF2α phosphorylation in the BAT of Ssu72 aKO mice than in Ssu72 WT mice (Supplementary Fig. 7b). Previous reports have demonstrated that IRE1α-Xbp1 pathway of UPR is required for adipogenesis and that activation of IRE1α-Xbp1 in a PKA-dependent manner promotes the transcription of *Ucp1* in brown adipocytes[45]. To assess whether Ssu72 depletion affected IRE1α-Xbp1 signaling pathway under cold exposure, we measured levels of spliced *Xbp1* induced by activation of IRE1α in BAT. Indeed, there was no significant difference in IRE1α activation between Ssu72 WT and aKO mice under RT or cold condition (Supplementary Fig. 7c). We also assessed expression levels of other eIF2α phosphatases such as GADD34, CReP, and PP1 in BAT of Ssu72 WT and aKO mice housed at RT. We found that only *Ppp1r15a* (*GADD34*) mRNA expression level was upregulated in Ssu72 aKO mice, but there was no difference in mRNA expression of *Ppp1r15b* (*CReP*) and *Ppp1ca* (*PP1*) between the genotypes (Supplementary Fig. 8a). The increased expression level of *Ppp1r15a* in aKO mice was probably due to compensation for hyperphosphorylation of eIF2α. Considering the feedback control of other eIF2α phosphatases in normal cells, we next explored effects of eIF2α phosphatases in adipocytes. Remarkably, data analysis of differential correlation score between each phosphatase gene and cell-type-enriched transcripts showed that only *Ssu72* was predominantly enriched in adipocytes compared to other eIF2α phosphatases (*GADD34, CReP, PP1*) (Supplementary Fig. 8b). This result suggests that the effect of Ssu72 phosphatase on eIF2α dephosphorylation in adipocytes might be greater than those of other eIF2α phosphatases.

To determine the potential interaction between Ssu72 and eIF2α, co-immunoprecipitation (Co-IP) assay was performed using cell lysates of 293T cells transfected with Myc-tagged Ssu72 (Myc-Ssu72) and HA-tagged eIF2α (HA-eIF2α). Interestingly, a complex formation of Ssu72 with eIF2α was found (Fig. 4f). This result was confirmed by in vitro binding assays using glutathione-S-transferase (GST) or GST-fused eIF2α (GST-eIF2α) and purified His-tagged Ssu72 (His-Ssu72). After His-Ssu72 protein was individually mixed with glutathione beads-bound GST or GST-eIF2α, results showed that His-Ssu72 bound to GST-eIF2α specifically as compared to control GST alone (Supplementary Fig. 9a, b). We further examined whether Ssu72 phosphatase could control eIF2α phosphorylation depending on Ssu72 phosphatase activity. As a negative control, we generated His-tagged Ssu72 phosphatase-dead mutant (His-Ssu72 C12S) and assessed the phosphatase activity of His-Ssu72 WT and C12S through p-nitrophenyl phosphate (pNPP) assay (Supplementary Fig. 9c). To directly investigate whether eIF2α could be dephosphorylated by active Ssu72, purified GST-eIF2α was phosphorylated by recombinant PERK. Phosphorylated GST-eIF2α (P-eIF2α[Ser51]) was then incubated with purified His-Ssu72 WT or His-Ssu72 C12S. Results clearly showed that only Ssu72 WT, but not phosphatase-dead Ssu72 C12S, dephosphorylated P-eIF2α[Ser51], indicating that Ssu72 could dephosphorylate P-eIF2α[Ser51] in vitro (Fig. 4g and Supplementary Fig. 9d).

We next asked whether expressing ectopic Ssu72 was sufficient to reduce eIF2α phosphorylation in vivo. To this end, we generated conditional transgenic (cTg) mice with *Rosa26[loxP-STOP-loxP]-HA*-tagged *Ssu72* gene (*Rosa26 HA-Ssu72*). These cTg mice were crossed with *Ssu72[f/f]; Adiponectin-Cre* (Ssu72 aKO) mice to generate *Ssu72[f/f]; Adiponectin-Cre; Rosa26 HA-Ssu72* mice (hereafter aKO;cTg), which expresses a single copy of ectopic *HA*-tagged *Ssu72* in a Cre-mediated adipocytes specific manner in the genetic background of Ssu72 aKO mice. In the absence of Cre recombinase, the generated cTg mouse line was unable to express ectopic *HA-Ssu72* gene nor to delete the endogenous *Ssu72* gene. Upon Cre recombination, ectopic *HA-Ssu72* cassette flanked by *loxP* sites was removed, *HA-Ssu72* became expressed, and endogenous *Ssu72* gene was deleted (Fig. 5a, b). Immunoblotting analysis revealed that ectopic HA-Ssu72 protein was expressed higher in BAT and iWAT than in eWAT from cTg mice (Fig. 5c). To assess sizes of adipocytes in Ssu72 WT, aKO, and aKO;cTg mice housed at RT, we performed H&E staining of BAT sections from 14-week-old mice. As expected, Ssu72 aKO mice exhibited white-like phenotype of BAT compared with WT mice. However, this morphological alteration was restored by adipose HA-Ssu72 expression in aKO;cTg mice (Fig. 5d). To further examine whether ectopic Ssu72 expression could reduce eIF2α phosphorylation in Ssu72-deficient BAT, immunohistochemistry staining and immunoblotting analysis were performed. We found that an increased eIF2α phosphorylation at serine 51 in Ssu72 aKO BAT was reduced by ectopic HA-Ssu72 expression in Ssu72 aKO;cTg BAT (Fig. 5e, f). Collectively, these data suggest that Ssu72 can act as eIF2α phosphatase in BAT.

## Ssu72-mediated translational control seems to be critical for BAT thermogenesis

Since phosphorylation of eIF2α inhibits global protein synthesis[19], we next assessed changes in translation between Ssu72 WT and Ssu72 aKO mice. To directly monitor mRNA translation in BAT, surface sensing of translation (SUnSET) assay, a nonradioactive method for detecting puromycin incorporated neosynthesized proteins[46,47], was performed. Immunoblotting analysis revealed that puromycin incorporation was remarkably blocked in BAT of Ssu72 aKO mice than in Ssu72 WT mice (Fig. 6a), indicating that general protein synthesis was inhibited by Ssu72 deficiency in BAT. Given that deficiency of Ssu72 caused cold intolerance in Ssu72 aKO mice during acute cold exposure (4 °C), we further investigated changes in the translation of major thermogenic factors in BAT. Of note, protein expression levels of PGC-1α, AMPKα,

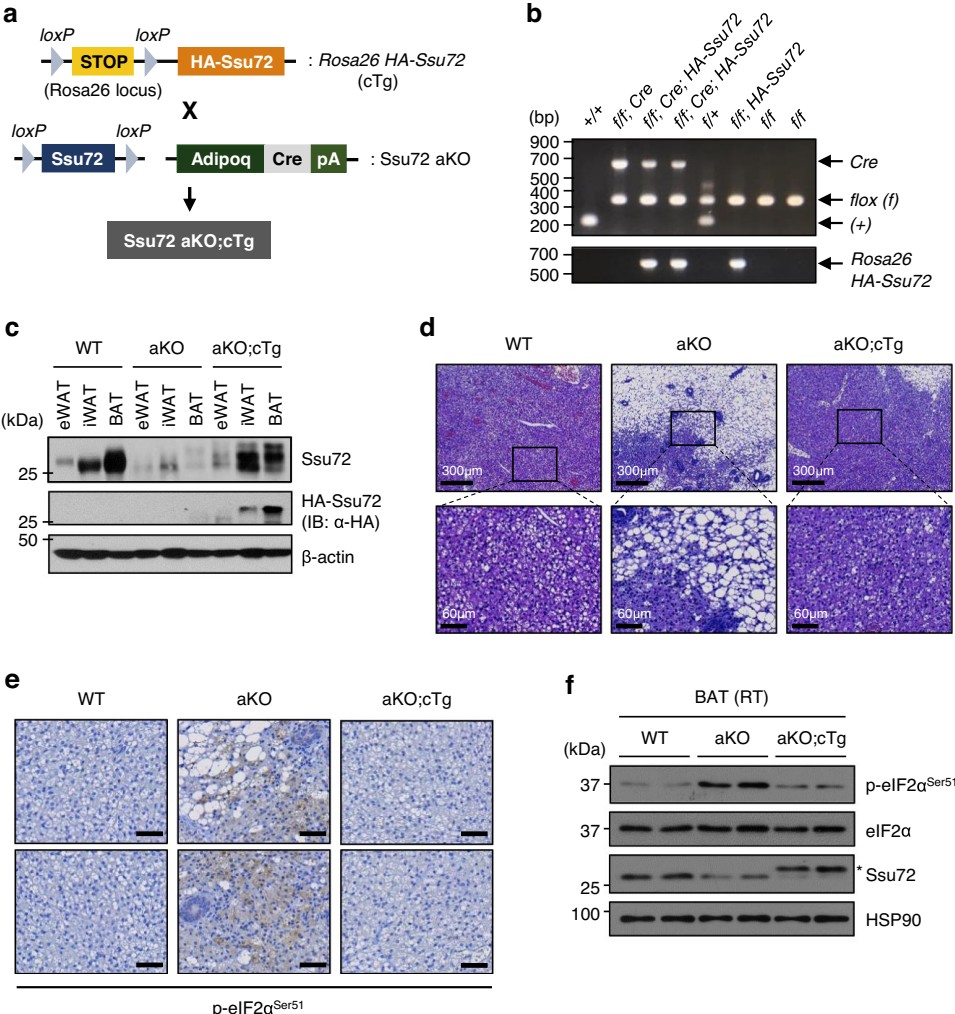

**Fig. 5 | Ectopic Ssu72 expression in adipose tissue restores morphological alteration of Ssu72-deficient BAT and reduces eIF2α phosphorylation in vivo.** **a** Schematic strategy for generating *Adiponectin-Cre*; Ssu72*flox/flox*; *Rosa26 HA-Ssu72* (Ssu72 aKO;cTg) conditional transgenic (cTg) mice. **b** PCR analysis with genomic DNA from mouse tail containing *floxed* (*f*), *Adiponectin-Cre* (*Cre*), and *Rosa26 HA-Ssu72* loci (*n* = 4–5 per group, biologically independent samples). **c** BAT extracts of 6-week-old WT, aKO and aKO;cTg mice were immunoblotted with anti-Ssu72, anti-HA (ectopic HA-Ssu72), and anti-GAPDH antibodies (*n* = 4 per group, biologically

independent samples). **d** H&E staining analysis of BATs from 14-week-old female WT, aKO, and aKO; cTg mice housed at RT (*n* = 4 per group, biologically independent samples). **e** Immunohistochemical analysis with α-P-eIF2α^Ser51 in BATs of WT, aKO, and aKO;cTg mice housed at RT (Scale bar, 50 μm) (*n* = 4 per group, biologically independent samples). **f** Western blots of BATs from WT, aKO, and aKO;cTg mice housed at RT. The asterisk marks the HA-Ssu72 (*n* = 2 mice per group). Source data are provided as a Source Data file.

and PKA were significantly lower in Ssu72 aKO BAT under basal condition (RT) and acute cold condition (4 °C) than in Ssu72 WT BAT (Fig. 6b). Moreover, in BAT of Ssu72 aKO mice, protein expression levels of UCP1 were significantly repressed during acute cold exposure, with a trend toward the decreased expression of these proteins at RT (Fig. 6b). To investigate whether protein synthesis was regulated by Ssu72 expression, we performed immunoblotting analyses for BATs of Ssu72 WT, aKO, and aKO;cTg mice housed at RT. Intriguingly, downregulated PGC-1α protein expression in BAT of Ssu72 aKO mice was completely restored by ectopic Ssu72 expression in BAT of aKO;cTg mice, along with dephosphorylated P-eIF2α^Ser51 (Fig. 6c). We also found that slightly reduced AMPKα protein expression in aKO mice was restored in aKO;cTg mice (Fig. 6c). We further assessed expression levels of thermogenic genes in BAT and found that mRNA levels of thermogenic genes (*Ucp1, Cidea, Dio2,* and *Ppargc1a*) in aKO;cTg mice were similar to those in WT mice (Supplementary Fig. 10a). The higher mRNA expression level of *Ppargc1a* in BAT of aKO mice than in WT mice (Supplementary Fig. 10a) was probably due to compensation for reduced protein expression of PGC-1α.

To further examine whether expression of ectopic HA-Ssu72 in BAT could rescue the thermogenic defect of Ssu72 aKO mice, 7-week-old Ssu72 WT, aKO, and aKO;cTg mice were challenged with acute cold exposure (4 °C). When observing behaviors of mice in response to acute cold, we found that the sluggish and shivering behavior of Ssu72 aKO mice after 4 h of cold exposure seemed to be restored by ectopic Ssu72 expression in aKO;cTg mice (Supplementary Movie 1 and Supplementary Fig. 10b). Morphological alterations that occurred in BAT of Ssu72 aKO mice were almost completely restored in BAT of aKO;cTg mice under severe cold condition (Fig. 6d). Remarkably, the severe cold intolerance of Ssu72 aKO mice was lethal, whereas cold susceptibility of aKO;cTg mice was similar to that of WT mice (Supplementary Fig. 10c). We further assessed UCP1 protein expression in BAT of 11-week-old cold-exposed mice and found that reduced UCP1 expression in Ssu72 aKO mice was restored by ectopic Ssu72 expression in aKO;cTg mice during cold exposure (Fig. 6e). Taken together, these results indicate that Ssu72 can mediate translation of key thermogenic factors in BAT under RT condition, which may affect cold tolerance of mice during acute cold exposure.

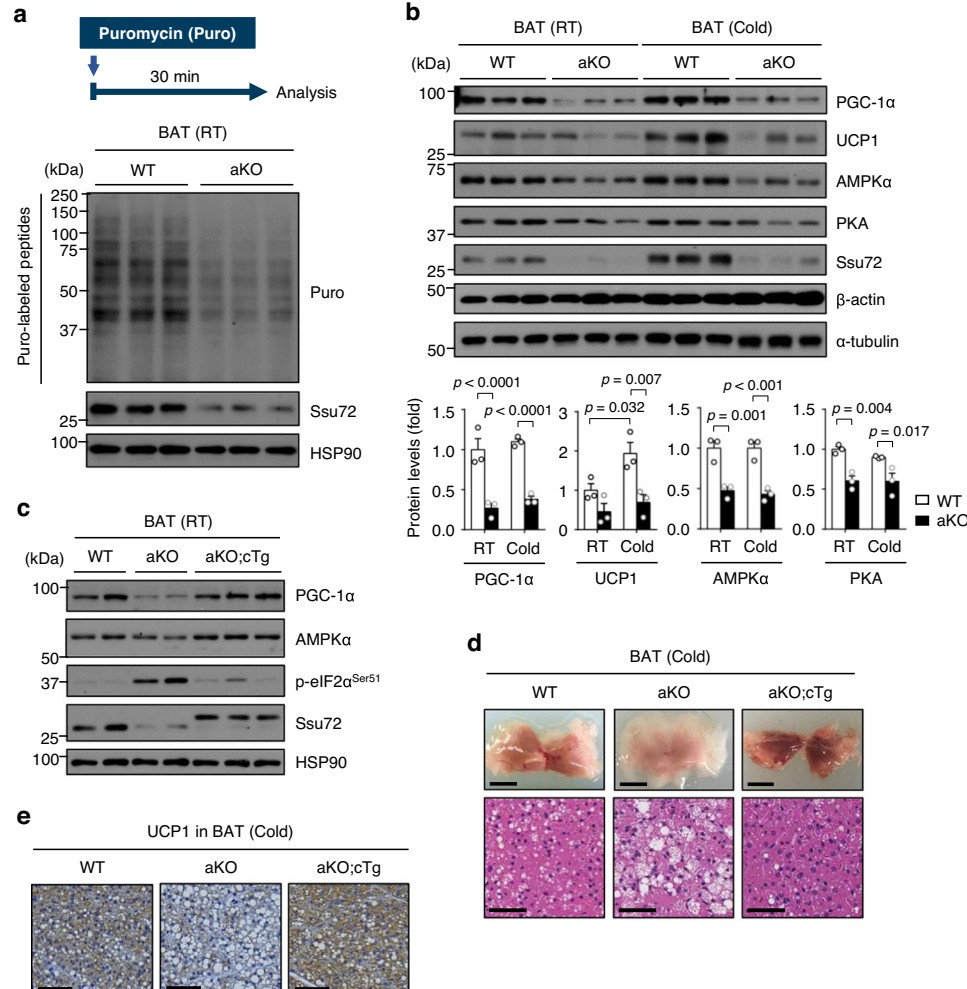

**Fig. 6 | Disturbances of translational control by Ssu72 depletion induces thermogenic defect in BAT. a** Representative images of WB-SUnSET analysis of BAT from 11-week-old WT and aKO male mice treated with low concentration of puromycin. Changes in protein synthesis were compared against vehicle treated cells and equal loading was confirmed with HSP90 ($n = 3$ mice per group).
**b** Western blots of thermogenic factors in BAT from 11-week-old Ssu72 WT and aKO male mice exposed to RT or acute cold (4 °C) for 4 h (upper panels) ($n = 3$ mice per group), with quantification of protein levels (bottom panels). Graph shows quantification of protein levels normalized to α-tubulin protein level using ImageJ. Data are presented as mean ± SEM. Statistical comparisons were made using two-way

ANOVA. **c** Western blots in BAT from 12-week-old Ssu72 WT, aKO and aKO;cTg female mice housed at RT ($n = 3$ mice per group). **d** Representative images (upper panels; scale bar, 5 mm) and H&E staining analysis (bottom panels; scale bar, 50 μm) of BATs from 8-week-old WT, aKO, and aKO;cTg male mice exposed to acute cold (4 °C) for 4 h ($n = 5$ per group, biologically independent samples).
**e** Immunohistochemical analysis with α-UCP1 in BATs of 11-week-old WT, aKO, and aKO;cTg female mice exposed to cold (4 °C) for 4 h (Scale bar, 60 μm) ($n = 3$ per group, biologically independent samples). Source data are provided as a Source Data file.

## Prolonged eIF2α phosphorylation by Ssu72 depletion reduces mitochondrial OXPHOS translation in BAT

Mitochondrial unfolded protein response (UPR<sup>mt</sup>) is activated by mitochondrial stress and multiple forms of mitochondrial defects, subsequently inducing nuclear transcriptional response to restore impaired mitochondrial proteome[48,49]. To address whether prolonged eIF2α phosphorylation by Ssu72 depletion was associated with mitochondrial dysfunction in BAT, we next measured levels of UPR<sup>mt</sup> transcripts as a marker of mitochondrial defects. We found that expression levels of UPR<sup>mt</sup> genes (*Hspa9, Lonp1, Hspd1*, and *Yme1l1*) in BAT of Ssu72 aKO mice were increased compared with those of Ssu72 WT mice (Supplementary Fig. 11a). A recent study has reported that prolonged UPR<sup>ER</sup> suppresses mitochondrial protease ClpP expression through eIF2α pathway[50]. Consistent with the report, mitochondrial protease ClpP expression was downregulated in Ssu72 aKO mice at both transcriptional and translational levels (Fig. 7a and Supplementary Fig. 11b). Furthermore, it has been reported that downregulation of ClpP in mammalian cells can attenuate mitochondrial OXPHOS

capacity[50,51], and ClpP-deficient mice show decreased mitoribosomal assembly in heart mitochondria, thereby affecting mitochondrial translation[52]. OXPHOS complex subunits are encoded on both nuclear and mitochondrial genomes[53]. To investigate whether regulation of ClpP expression in Ssu72-deficient BAT could inhibit mitochondrial OXPHOS capacity, we examined expression levels of OXPHOS subunits encoded by nuclear DNA, including NDUFB8 (complex I, CI), SDHB (complex II, CII), UQCRC2 (complex III, CIII), COX4 (complex IV, CIV), and ATP5A (complex V, CV), in BAT of Ssu72 WT and aKO mice housed at RT. It was found that mRNA levels of these nuclear-encoded OXPHOS genes were not significantly different between genotypes (Fig. 7b). Surprisingly, protein expression levels of CI, CII, and CIV subunits were markedly reduced in BAT of Ssu72 KO mice than in Ssu72 WT mice (Fig. 7c), supporting the notion that downregulation of ClpP expression in Ssu72-deficient BAT could inhibit cytosolic translation of OXPHOS subunits. We further investigated expression levels of mitochondrial-encoded OXPHOS subunits. We found that there was no significant difference at transcriptional levels (Fig. 7d). Notably,

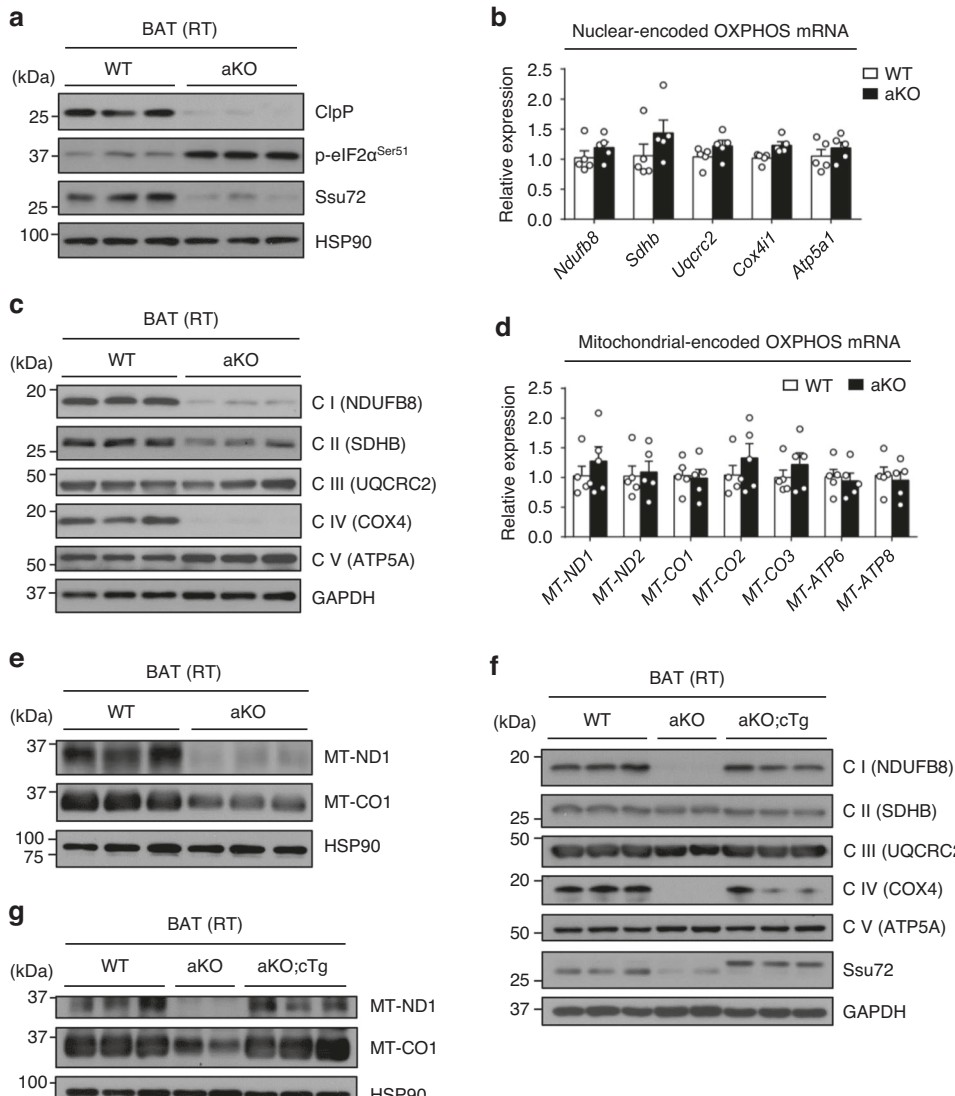

**Fig. 7 | Ssu72 depletion in BAT affects translation of OXPHOS subunits.**
**a** Western blots of ClpP and phosphorylated eIF2α (P-eIF2α^Ser51) in BAT from Ssu72 WT and aKO mice housed at RT (*n* = 3 mice per group). **b** Relative mRNA expression levels of OXPHOS complex subunits encoded by nuclear DNA in BAT from Ssu72 WT and aKO mice housed at RT (*n* = 5 mice per group). Data are presented as mean ± SEM. Statistical comparisons were made using two-tailed Student's *t* test. **c** Western blots of mitochondrial OXPHOS complex subunits (CI-V) encoded by nuclear DNA in BAT from Ssu72 WT and aKO mice housed at RT (*n* = 3 mice per group). **d** Relative mRNA expression levels of OXPHOS complex subunits encoded by mitochondrial DNA (Complex I, IV, and V) in BATs from Ssu72 WT and aKO mice

housed at RT (*n* = 5 mice per group). Data are presented as mean ± SEM. Statistical comparisons were made using two-tailed Student's *t* test. **e** Western blots of MT-ND1 and MT-CO1 in BAT from Ssu72 WT and aKO mice housed at RT (*n* = 3 mice per group). **f** Western blots of OXPHOS complex subunits (CI-V) encoded by nuclear DNA in BAT from 12-week-old Ssu72 WT, aKO, and aKO;cTg female mice housed at RT (*n* = 3 mice per group). **g** Western blots of MT-ND1 and MT-CO1 in BAT from 12-week-old Ssu72 WT, aKO and aKO;cTg female mice housed at RT (*n* = 3 mice per group). For **a**–**e**, 11-week-old male Ssu72 WT and aKO mice were used. Source data are provided as a Source Data file.

protein expression levels of MT-ND1 and MT-CO1 were downregulated in Ssu72-deficient BAT (Fig. 7e). To validate the effect of Ssu72 on the translation of OXPHOS complex, we examined whether ectopic Ssu72 expression could restore defects in mitochondrial translation in Ssu72-deficient BAT. We found that reduced protein expression levels of OXPHOS subunits in BAT of Ssu72 aKO mice were restored by ectopic HA-Ssu72 expression in BAT of aKO;cTg mice (Fig. 7f, g). Overall, we found that prolonged eIF2α phosphorylation by Ssu72 depletion in BAT could affect the expression of key functional factors while attenuating global mRNA translation and decrease ClpP expression, thereby reducing mitochondrial OXPHOS capacity and thermogenesis (Supplementary Fig. 11c). These findings suggest that cytosolic translation regulated by Ssu72 is critical for mitochondrial function and BAT thermogenesis during cold exposure.

## Discussion

The prevalence of metabolic diseases is increasing, and defective adaptive thermogenesis is associated with the progression of metabolic diseases in humans[54–56]. Therefore, identifying molecular mechanisms that increase BAT activity will be interesting future avenues. In this study, we demonstrate that Ssu72 phosphatase plays a critical role in BAT thermogenesis. The findings of this study revealed that protein expression of Ssu72 was specifically high in BAT, and its expression was markedly increased upon acute cold challenge. We also found that cold tolerance was dramatically reduced in mice lacking adipocyte Ssu72 (Ssu72 aKO) upon acute cold (4 °C) exposure, and that defect could be restored by ectopic Ssu72 expression in adipose tissues. These findings suggest that Ssu72 expression in adipose tissue can enhance the thermogenic activity of BAT.

During thermogenic adaptation, BAT undergoes complex catabolic pathways for respiration, while simultaneously challenging demands of synthetic anabolic processes such as increased protein synthesis and de novo lipogenesis[18,57]. Indeed, not only transcriptional regulation but also post-transcriptional regulation such as RNA processing and translation are highly induced during BAT activation by acute cold exposure[58], suggesting that mRNA translation is important for thermogenic adaptation of BAT in response to acute cold. Phosphorylation of eIF2α is a well-known mechanism for the regulation of mRNA translation initiation, which inhibits global protein synthesis in eukaryotic cells[22]. In an obese state, eIF2α is phosphorylated to cope with chronic ER stress and to maintain protein homeostasis[59]. Although recent evidence has revealed that proteasomal protein quality control is essential for thermogenesis upon chronic exposure to cold or excess nutrients[60], little is known about how BAT responds to the high demand for protein synthesis for thermogenic functions during acute cold exposure. Here, we found that Ssu72 could act as an eIF2α phosphatase and mediates protein synthesis of key thermogenic effectors in BAT. Loss of Ssu72 increased phosphorylation of eIF2α at Ser51 in BAT, whereas adipocyte-specific Ssu72 expression dramatically downregulated phosphorylation of eIF2α. Our findings provide evidence for a role of Ssu72 in protein synthesis upon mild (RT) and severe cold (4 °C) exposure, thereby affecting thermogenic activity of BAT. Unexpectedly, transcriptional control of thermogenic genes was also shown in Ssu72-deficient BAT during severe cold exposure. Several studies have shown that impaired mitochondrial respiratory capacity can result in decreased expression of thermogenic genes in BAT[61–63], implying that BAT mitochondria can sense its respiratory capacity and communicates with the nucleus to regulate the transcription of genes[64]. Since severe cold stimulation requires high respiratory capacity in brown adipocytes and enhances the thermogenic gene program[16,65], expression levels of thermogenic genes are increased within a few hours of severe cold exposure with normal respiratory capacity. However, thermogenic gene expression in Ssu72-deificent brown adipocytes might not be increased in response to severe cold due to a secondary effect of reduced mitochondrial respiratory capacity.

Why and how does Ssu72 phosphatase predominantly control the function of BAT relative to WAT? BAT has a high density of mitochondria with a higher amount of Ucp1 than WAT[66]. It should be noted that Ssu72 plays a substantial role in mitochondria-rich adipocytes. Much evidence from our study clearly shows that Ssu72 deficiency causes severe mitochondrial dysfunction and affects thermogenic adaptation of BAT. Another interesting implication is that BAT whitening that occurs with sustained high-fat food intake is closely related to mitochondrial dysfunction[67]. Of note, in mice housed at room temperature and fed a normal chow diet, Ssu72 deficiency resulted in BAT whitening with large and unilocular lipid droplets. The white-like phenotype of Ssu72-deficient BAT was partially restored by the expression of ectopic Ssu72 in vivo. In fact, housing mice at room temperature (20 °C–24 °C) is considered mild cold exposure, as about 30% of resting energy expenditure is used for thermoregulation at this temperature[43]. Upon adaptation to thermoneutrality (TN, 30 °C), BAT seems to adopt a white-like characteristics[68]. Hence, Ssu72 serves as a key regulator to prevent brown-to-white adipose tissue conversion, allowing BAT to maintain its thermogenic activity under cold condition. Inguinal WAT (iWAT) also undergoes adaptive and dynamic changes in response to cold. Chronic severe cold exposure or β3-adrenergic receptor agonists can induce the generation of mitochondrial-rich thermogenic beige adipocytes[69]. At room temperature, Ssu72 expression in iWAT was lower than that in BAT. However, Ssu72 expression in iWAT was increased after exposure to cold (4 °C) for 8–12 h. This finding may point to an additional role for Ssu72 in iWAT. Further studies are needed to investigate whether Ssu72 affects the browning or thermogenic function of iWAT during chronic exposure to severe cold.

It has been reported that eIF2α kinase PERK can coordinate mitochondrial molecular quality control in response to ER stress[70]. Under ER and nutrient stress conditions, PERK-eIF2α pathway promotes respiratory supercomplex assembly through supercomplex assembly factor 1 (SCAF1); however, this pathway is insufficient to form mitochondrial cristae[71]. A recent study has shown that PERK is required for cristae formation though the PERK-GABPα pathway during brown adipocyte differentiation[72]. In addition, PERK promotes cristae formation by enhancing mitochondrial import of MIC19, a key organizer of cristae formation, and the induction of MIC19 is independent of the canonical PERK-eIF2α pathway[73]. Considering the direct effect of Ssu72 on eIF2α dephosphorylation, mitochondrial dysfunction in Ssu72 aKO mice is more affected by P-eIF2α-dependent translational control than by PERK-dependent mechanisms.

An important question emerging from our study is how persistent translational control causes mitochondrial dysfunction. An earlier study using advanced ribosome profiling approach has revealed that inhibition of cytosolic translation affects mitochondrial translation and that synchronization of translation is controlled by unidirectional communication from cytosol to mitochondria[74]. In addition, a flux of nuclear-encoded molecules from cytosol to mitochondria triggers translation of mitochondrial-encoded OXPHOS subunits[75]. Indeed, in BAT of Ssu72 aKO mice, cytosolic translation was globally reduced and the translation of nuclear-encoded OXPHOS subunits was attenuated. Consistent with previous reports[50,52], we also found that prolonged eIF2α phosphorylation by Ssu72 depletion downregulated ClpP expression, possibly affecting mitochondrial translation of OXPHOS subunits. In this regard, we believe that Ssu72 is also associated with mitochondrial translation. We found that the translation of mitochondrial-encoded OXPHOS subunits, MT-ND1 and MT-CO1, was attenuated in Ssu72-deficient BAT. Our results suggest that persistent control of translation through eIF2α phosphorylation by Ssu72 deficiency can reduce cytosolic translation and even mitochondrial translation of OXPHOS subunits.

In summary, Ssu72 is a cold stress-responsive protein phosphatase that mediates cytosolic translation and mitochondrial oxidative phosphorylation, which is required for thermogenesis in brown adipocytes (Fig. 8). Our results demonstrate that Ssu72 phosphatase affects protein synthesis and thermogenic capacity of BAT upon cold exposure. Our results also demonstrate that Ssu72-mediated mRNA translation by regulating eIF2α phosphorylation can induce the expression of key functional proteins and increase mitochondrial OXPHOS capacity, fueling thermogenic activation in BAT. Beyond the major role of BAT in heat production, recent studies using murine models have shown that BAT transplantation improves systemic energy expenditure and protects mice from obesity, diabetes, and liver steatosis[76–78]. Furthermore, targeting and enhancing BAT activity could be a promising therapeutic tool to treat metabolic diseases in humans[12,79,80]. In this context, our study raises the strong possibility that Ssu72 is a potential therapeutic target for several metabolic diseases including obesity-related diseases and non-alcoholic fatty liver disease (NAFLD). It is of great interest to investigate metabolic benefits of increasing Ssu72 expression and/or activity in patients.

## Methods

### Mice

Male C57BL/6J (WT) mice were obtained from Laboratory Animal Research Center (LARC) of Sungkyunkwan University School of Medicine. *Ssu72*[flox/flox] mice (Ssu72 WT) were bred with *Adiponectin-Cre* mice to generate mature adipocyte-specific *Ssu72* knockout (*Ssu72*[flox/flox]; *Adiponectin-Cre*) mice (Ssu72 aKO). We also generated conditional transgenic mice (cTg) with gene encoding ectopic *HA*-tagged *Ssu72* in the *Rosa26* locus (*Rosa26*[loxp-STOP-loxP]-*HA-Ssu72*). These transgenic mice

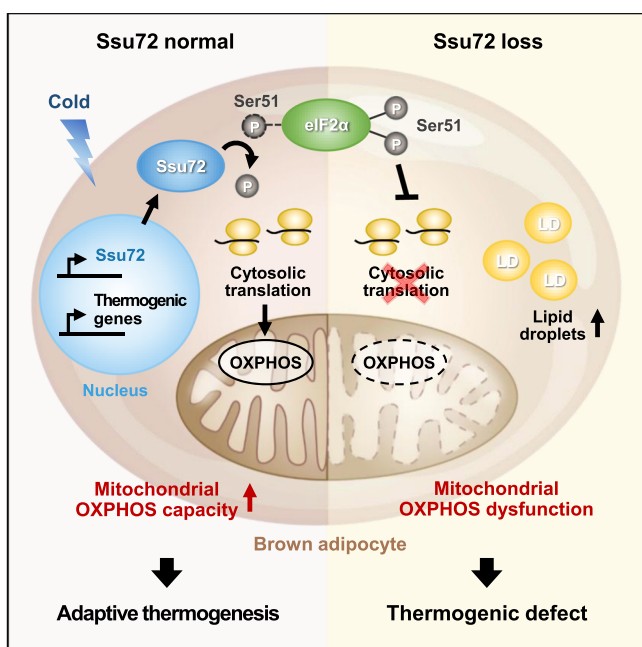

**Fig. 8 | A proposed model for the role of Ssu72 in translation required for thermogenic adaptation of BAT.** Ssu72, whose expression is increased in response to acute cold, acts as an eIF2α phosphatase that dephosphorylates P-eIF2α$^{Ser51}$ and induces cytosolic translation of key thermogenic effectors such as OXPHOS. Through these mechanisms, mitochondria with normal oxidative capacity further activate adaptive thermogenesis in brown adipocytes. In Ssu72-deficient brown adipocytes, cytosolic translation is attenuated by hyperphosphorylation of eIF2α, resulting in mitochondrial dysfunction and thermogenic defect.

(*Rosa26 HA-Ssu72*) were then crossed with Ssu72 aKO mice to generate aKO;cTg mice. All mice were maintained on a C57BL/6J background. Genotyping was performed by PCR analysis, and PCR primers for genotyping are as follows: *Ssu72 flox* (forward), 5′-TCAAAGC ATGATTGAGAGCAGCAG-3′; *Ssu72 flox* (reverse), 5′-GTGA-TAGGCAAGCAGGTGTGAG-3′; *Cre* (forward), 5′-GTCGATGCAACGAGT GATGA-3′; *Cre* (reverse), 5′-TCATCAGCTACACCAGAGAC-3′; *Rosa26 HA-Ssu72* (forward), 5′-AAAGTCGCTCTGAGTTGTTAT-3′; *Rosa26 HA-Ssu72* (reverse), 5′- GGAGCGGGAGAAATGGATATG-3′.

Mice were maintained under temperature- (23 °C) and humidity-controlled (40–60%) conditions with free access to food (normal chow diet; LabDiet, #5053) and water, on a 12-h light/12-h dark cycle. All mice are age and sex matched for individual experiments. Both male and female mice were used in the experiments, and no sex-specific phenotype was observed in vivo. For acute cold challenge experiments, mice had *ad libitum* access to water, although food was removed for a short period of time when animals were placed at 4 °C. During cold stress, a temperature probe (JEUNG DO Bio & Plant, #JD-DT-08g) was implanted into the anus of each mouse every hour. All mice used in the experiments were euthanized with $CO_2$ inhalation until breathing and heartbeat stopped. All animal experiments were conducted in accordance with guidelines of the Institutional Animal Care and Use Committee (IACUC) of Sungkyunkwan University School of Medicine (SUSM), which is accredited by the Association for Assessment and Accreditation of Laboratory Animal Care International (AAALAC International) and abides by the Institute of Laboratory Animal Resources (ILAR) guidelines.

### Gene expression Omnibus (GEO) dataSets analysis
For comparison of *Ssu72* mRNA expression in human adipocytes between lean and obese individuals, each gene expression datasets were obtained from GEO database. Relative Ssu72 expression value in

human subcutaneous adipocytes from non-diabetic obese individuals ($n = 19$) was compared with non-diabetic lean individuals ($n = 20$) (GEO accession number: GSE2508).

### Cell culture
293T cells (ATCC, #CRL-3216) were cultured in Dulbecco's Modified Eagle Medium (DMEM) (GenDEPOT) supplemented with 10% FBS (Gibco) and 1% penicillin/streptomycin. These cells were maintained at 37 °C in a humidified atmosphere with 5% $CO_2$.

### RNA isolation from adipose tissues
Total RNA was isolated from adipose tissues using QIAzol Lysis Reagent (QIAGEN, #79306). Adipose tissues (BAT, iWAT, eWAT) were placed in 1 ml QIAzol Lysis Reagent, homogenized with a Polytron Homogenizer PT 1300D (Kinematica AG, Switzerland), and centrifuged at 12,000×$g$ for 10 min (4 °C) to remove and discard the fatty layer. Chloroform was then added to cleared homogenized samples. The samples were centrifuged at 12,000×$g$ for 15 min (4 °C) and aqueous phases were obtained. Total RNA was precipitated with the addition of isopropanol and centrifugation at 12,000×$g$ for 10 min (4 °C). The supernatant was removed, and RNA pellet was washed with ethanol. The pellet was resuspended with RNase-free water and the concentration was measured using a NanoDrop spectrophotometer.

### Quantitative real-time PCR (qRT-PCR)
cDNA templates were synthesized with the cDNA synthesis kit (Applied Biological Materials Inc. (Abm), #G236) using random primers. The cDNA was then subjected to PCR amplification using 2X PCR Taq MasterMix (Abm) and gene-specific primers. Quantitative real-time PCR analysis was performed with PowerUP SYBR Green Master Mix (Applied Biosystems) reagent. Reactions were performed in triplicate. Relative gene expression was calculated using the comparative Ct ($2^{-\Delta\Delta Ct}$) method[81], where values were normalized to a 5S or 18S rRNA gene expression. The mRNA expression of candidate genes was determined by QuantStudio 6 Flex Real-Time PCR (Life technologies). All primer sequences for qRT-PCR are provided in Supplementary Table 1.

Mitochondrial DNA (mtDNA) copy number was measured by qRT-PCR. Total genomic DNA was isolated from BAT tissues using a DNeasy 96 Blood & Tissue Kit (QIAGEN, #69504). Then, 100 ng total DNA was used for mtDNA quantification. The copy number was normalized to nuclear DNA. The primer sequences are provided in Supplementary Table 1.

### Western blotting and Coomassie brilliant blue staining
Protein lysates were prepared from tissues or cells using RIPA buffer (20 mM pH 7.4 Tris-HCl, 150 mM NaCl, 1% Triton X-100, 0.1% SDS, 1 mM EGTA, 1 mM Phenylmethylsulfonyl fluoride (PMSF), 1 mM $Na_3VO_4$ and 1X Protease inhibitor cocktail (PIC)). For tissue lysate preparation, tissues were homogenized with a Polytron Homogenizer PT 1300D (Kinematica AG, Switzerland) in cold RIPA buffer. For cell lysis, cells were resuspended in cold RIPA buffer and homogenized by passing cells through a syringe tip. Equal amounts of protein lysates as quantified by Bradford assay were separated by sodium dodecyl sulfate-polyacrylamide gel electrophoresis (SDS-PAGE) and then transferred to nitrocellulose membranes using a Mini Trans-Blot Electrophoretic Transfer Cell (Bio-Rad). Membranes were blocked with Tris-Buffered Saline with 0.1% tween 20 (TBS-T) containing 5% (w/v) skim milk for 1 h at room temperature (RT) and incubated with primary antibodies in 5% skim milk or 5% BSA for 3 h at RT or overnight at 4 °C. These membranes were washed for 30 min with TBS-T, subsequently incubated with horseradish peroxidase (HRP)-conjugated secondary antibodies in 5% skim milk for 2 h at RT, and then washed for one hour with TBS-T. Blotted proteins were detected using ECL solution (AbFrontier) and exposed on Medical X-ray film blue (AGFA).

The following primary antibodies were used (at the indicated dilutions): anti-Ssu72 (Cell signaling Technology (CST), #12816, 1:2000), anti-PP1α (CST, #2582, 1:750), anti-CDC25B (Thermo Fisher Scientific, #PA5-17759, 1:1000), anti-PTEN (CST, #9188, 1:1000), anti-PP2A-B56-α (Santa Cruz Biotechnology (SCBT), #sc-271311, 1:1000), anti-β-actin (Sigma-Aldrich, #A2066, 1:3000), anti-UCP1 (SCBT, #sc-6529, 1:750), anti-GAPDH (CST, #2118, 1:3000), anti-PGC1α (SCBT, #sc-13067, 1:750), anti-HSP90 (SCBT, #sc-13119, 1:1000), anti-phospho-eIF2α (Ser51) (CST, #3398, 1:1000), anti-eIF2α (CST, #5324, 1:2000), anti-ATF4 (SCBT, #sc-390063, 1:1000), anti-CHOP (CST, #2895, 1:750), anti-phospho-PERK (Thr980) (Thermo Fisher Scientific, #MA5-15033, 1:1000), anti-phospho-IRE1α (Ser724) (Thermo Fisher Scientific, #PA1-16927, 1:1000), anti-ATF-6 (CST, #65880, 1:1000), anti-Myc-Tag (CST, #2276, 1:1500), anti-HA-Tag (SCBT, #sc-7392, 1:1000), anti-HA-Tag (CST, #3724, 1:2000), anti-β-actin (CST, #3700, 1:3000), anti-GST (SCBT, #sc-138, 1:2000), anti-puromycin (DSHB, #PMY-2A4, 1:200), anti-AMPKα (CST, #5831, 1:2000), anti-PKA C alpha (GeneTex, #GTX104934, 1:1000), anti-ClpP (SCBT, #sc-271284, 1:1000), anti-total OXPHOS Rodent WB Antibody Cocktail (Abcam, #ab110413, 1:500), anti-COX IV (CST, #4850, 1:1000), anti-MT-ND1 (Abcam, #ab181848, 1:1000). The following secondary antibodies were used (at the indicated dilutions): anti-Rabbit IgG(H + L)-HRP (GenDEPOT, #SA002-500, 1:7000), anti-Mouse IgG(H + L)-HRP (GenDEPOT, #SA001-500, 1:7000), anti-Goat IgG(H + L)-HRP (GenDEPOT, #SA007-500, 1:7000). A list of all antibodies used in this study is provided in Supplementary Table 2.

For Coomassie brilliant blue staining, SDS-PAGE gels were stained for 1 h using Coomassie staining buffer (0.25% Coomassie Brilliant Blue R-250, 50% methanol, 10% acetic acid, 40% distilled water) and destained with a destaining buffer (50% methanol, 10% acetic acid, 40% distilled water) overnight at RT.

### Metabolic mouse studies
Metabolic studies were conducted at Soonchunhyang University under an approved SCH-IACUC protocol. Energy expenditure and associated experiments were measured using a Phenomaster (TSE systems) at Soonchunhyang Biomedical Research Core-facility of Korea Basic Science Institute (KBSI). Oxygen consumption (VO₂) and CO₂ release rates (VCO₂) were measured every 24 min. Energy expenditure and Respiratory exchange rate (RER) were calculated by the ratio between VO₂ and VCO₂. Food intake was automatically monitored by Phenomaster food measurement module and locomotor activity was obtained by counting the number of the $x$-axis and $y$-axis beam breaks. Ssu72 WT and aKO mice were injected intraperitoneally with a β₃-adrenergic receptor-specific agonist CL-316,243 (Sigma-Aldrich, #5976) at a dose of 10 mg/kg to examine their responses to adrenergic stimulation, and VO₂ was measured every 2 min.

### Transmission electron microscopy (TEM) analysis
BAT was fixed with 2% glutaraldehyde and 2% paraformaldehyde in 0.1 M sodium cacodylate buffer (pH 7.4) for 48 h. After dehydration in a graded acetone series, tissues were embedded in Spurr resin. Sections were cut on a Leica UCT ultramicrotome and placed onto Cu girds. Sections were post-stained with uranyl acetate and lead citrate, and then examined with a Hitachi H600AB transmission electron microscope at 75 kV. Three mice were analyzed for each genotype. The size of mitochondria was quantified using ImageJ software v.1.52a (NIH, https://imagej.nih.gov/ij/). Mitochondria with disrupted cristae and total mitochondria were counted from each image and then expressed as % cristae disruption (mitochondria with disrupted cristae over total mitochondria).

### Histological analysis and Immunohistochemistry (IHC) staining
Tissue samples (BAT, iWAT, and eWAT) were fixed in 10% formalin solution (Sigma-Aldrich, #HT501320) for 24–36 h at 4 °C, embedded in paraffin, cut at 5 μm (BAT) and 6 μm (eWAT and iWAT) thick sections, and then stained with hematoxylin and eosin (H&E). For immunohistochemistry, deparaffinization and dehydration of tissue paraffin sections were conducted with xylene and ethanol. Heat-induced antigen retrieval was performed by boiling a section in 10 mM citric acid buffer (pH 6.0) for 15 min at 95–100 °C. Slides were incubated with 3% hydrogen peroxidase to block the endogenous peroxidase activity. After washing with TBS-T buffer, sections were blocked with goat serum (1.5% blocking solution) for 1 h at room temperature followed by incubation with anti-eIF2α^Ser51 (Cell Signaling Technology, #3398) antibody diluted 1:150 or anti-UCP1 (Abcam, #ab10983) antibody diluted 1:500 at 4 °C overnight. These slides were incubated with biotinylated goat anti-rabbit IgG secondary antibody (Vector Laboratories, #PK-6101) diluted 1 drop (50 μl) to 10 ml blocking solution at room temperature for 1 h. Labeling was then visualized with 3,3'-diaminobenzidine (DAB). Nuclei were stained with hematoxylin. Slides were scanned with a MoticEasyScan Pro 6 (Motic) scanner and scanned images were viewed with Aperio ImageScope software v.12.4.3.5008 (Leica Biosystems).

### Immunofluorescence (IF) staining
For the preparation of frozen sections, BAT tissues were fixed with 4% paraformaldehyde (PFA) (Sigma-Aldrich, #P6148) at 4 °C for 24 h and washed with ice cold PBS. Fixed tissues were placed in 15% sucrose (Sigma-Aldrich, #S9378) in PBS at 4 °C for 6 h and then incubated in 30% sucrose in PBS at 4 °C overnight using a tube rotator. Tissues were embedded in OCT embedding matrix (Sakura Finetek, #4583) and stored at −70 °C. For immunofluorescence (IF) staining, frozen sections were incubated with 0.3% Triton X-100 in PBS for 30 min, washed with PBS, and blocked with PBS containing 5% goat serum and 0.1% Triton X-100 at RT for 40 min. These sections were then incubated with anti-eIF2α^Ser51 (Cell Signaling Technology, #3398) antibody diluted 1:200 at 4 °C overnight followed by incubation with Alexa Fluor 568-conjugated anti-rabbit IgG secondary antibody (Thermo Fisher Scientific) diluted 1:300 at RT for 2 h. Nuclei were stained with DAPI (4′,6-diamidino-2-phenylindole) using VECTASHIELD Antifade Mounting Medium with DAPI (Vector Laboratories, #H-1200). Images were taken with an Axio Imager microscope (ZEISS).

### RNA sequencing (RNA-Seq) and data analysis
RNA-Seq libraries were prepared with a TruSeq Stranded mRNA Library Prep kit (Illumina) using purified RNAs isolated from BAT of Ssu72 WT and aKO mice, including three independent samples. Transcriptomic sequencing was performed on a Novaseq platform using the standard sequencing protocol. In total, 50–65 million base pair (bp) reads were generated per sample. An initial sequence-level quality assessment was performed using FastQC (v0. 11. 9). RNA-Seq reads were then mapped to the mouse mm10 reference genome using Tophat (v2.0.13). Differential gene expression analysis was performed using cuffdiff [82] package (v2.2.0) (http://cole-trapnell-lab.github.io/cufflinks/releases/v2.2.0/). Raw read counts were used and modeled based on a negative binomial distribution. Genes were considered to be differentially expressed if they met the following criteria: (1) expression changes of 2-fold or greater between means of both Ssu72 WT and aKO samples, (2) $p$ value was less than 0.05, and (3) FDR was less than 0.1. Data were obtained from three independent experiments and processed using DAVID Bioinformatics Resources 6.7 software for Gene ontology, BIOCARTA, KEGG pathway, and gene set enrichment analyses. List of differentially expressed genes (DEGs) in BAT between Ssu72 WT and aKO mice from RNA-seq data is provided in Supplementary Data 1.

### Gene set enrichment analysis (GSEA)
Gene set enrichment analysis (GSEA) was performed with software version 4.1.0 (Broad Institute and University of California, https://www.

gsea-msigdb.org/gsea/index.jsp). Prior to analysis, a ranked list was computationally calculated with each gene based on $\log_2$ fold change. Gene sets were defined using the molecular signature database (MSigDB) curated from BIOCARTA, Kyoto Encyclopedia of Genes and Genomes (KEGG), REACTOME, and Pathway Interaction Database (PID). Gene sets with an FDR < 0.25 and a normal $p$ value <0.05 were considered statistically significant.

### Recombinant protein purification

To generate His-Ssu72 phosphatase-dead mutant (His-Ssu72 C12S), PCR-based site-directed mutagenesis was performed using a Muta-Direct™ Site Directed Mutagenesis Kit (INtRON, #15071). His-tagged recombinant proteins (His-Ssu72 WT and His-Ssu72 C12S mutant) were expressed in *Escherichia coli* (*E. coli*) strain BL21 competent cell (Enzynomics, #CP110) by overnight induction at 18 °C with 0.2 mM isopropyl-β-D-1-thiogalactopyranoside (IPTG). Pelleted bacteria were resuspended in a lysis buffer (300 mM NaCl, 20 mM Sodium phosphate (pH 8.0), 20 mM Imidazole, 1% Triton X-100, and 10% glycerol) supplemented with 1 mM PMSF and 1 mM dithiothreitol (DTT)). After sonication, lysates were centrifuged at 48,384×$g$ for 30 min at 4 °C. The supernatant of the lysate was incubated with 50% slurry of Ni-NTA beads (QIAGEN) for 3 h at 4 °C. Immobilized His proteins-Ni-NTA were then loaded onto a column and washed twice with a wash buffer (300 mM NaCl, 20 mM Sodium phosphate (pH 8.0), and 20 mM Imidazole). His proteins were then eluted with an elution buffer containing a high concentration of imidazole (300 mM NaCl, 20 mM Sodium phosphate (pH 8.0), and 300 mM Imidazole). Purified His proteins were concentrated using centrifugal concentrators (Sartorius).

For GST-tagged proteins purification, pGEX-KG (GST) and pGEX-KG-EIF2S1 (GST-eIF2α) were expressed in BL21 competent cell by overnight induction at 18 °C with 0.5 mM IPTG. Briefly, pelleted bacteria were resuspended with STE buffer (10 mM Tris-HCl (pH 8.0), 150 mM NaCl, and 1 mM EDTA) and lysed in STE buffer containing 100 μg/ml lysozyme, 0.5 mM PMSF, and 1X PIC. After sonication, lysates were centrifuged at 20,442×$g$ for 30 min at 4 °C. The supernatant was incubated with 50% slurry of Glutathione Sepharose 4B beads (Cytiva) overnight at 4 °C. These beads were then washed three times with cold 1X PBS. All purified proteins were separated by SDS-PAGE and then analyzed by Coomassie brilliant blue staining.

### In vitro binding assay and in vitro kinase/phosphatase assay

For in vitro binding assay, bead bound GST or GST-eIF2α proteins (2 μg) were incubated with His-Ssu72 WT (2 μg) in binding buffer (20 mM Tris-HCl (pH 7.4), 100 mM NaCl, 0.5% NP-40, 1 mM EDTA, and 1 mM DTT) with glutathione beads for 3 h at 4 °C. These beads were then washed three times with a wash buffer (20 mM Tris-HCl (pH 7.4), 150 mM NaCl, 1% NP-40, 1 mM EDTA, and 1 mM DTT). Bound proteins were eluted with an SDS sample buffer and then separated by SDS-PAGE.

For in vitro kinase/phosphatase assay, bead bound 2 μg GST or 2 μg GST-eIF2α protein was incubated with 0.5 μg recombinant PERK protein (Abcam, #ab101115) in kinase buffer (100 mM MOPS/NaOH (pH 7.2), 20 mM MgCl₂, 1 mM DTT and 1 mM PMSF) containing 1 mM adenosine 5′-triphosphate (ATP) (Sigma-Aldrich) at 37 °C for 1 h. Phosphorylated GST-eIF2α proteins were washed three times with phosphatase buffer (50 mM Tris-HCl (pH 7.4), 20 mM MgCl₂, 0.2 mM EDTA, and 0.2 mM EGTA), and incubated with purified His-Ssu72 WT or His-Ssu72 C12S mutant proteins at 37 °C for 90 min. The reaction was terminated by the addition of SDS sample buffer. Phosphorylation of eIF2α was then analyzed by western blotting.

### Co-immunoprecipitation (Co-IP) assay

293T cells were transfected with pCMV-Myc (Myc), pCMV-Myc-Ssu72 (Myc-Ssu72), and pCMV3-HA-EIF2S1 (HA-eIF2α; Sino Biological, #HG13107-NY). At 48 h after transfection, cells were washed twice with 1X PBS and harvested. Harvested cells were lysed using an IP lysis buffer (20 mM Tris-HCl (pH 7.4), 150 mM NaCl, 1% Triton X-100, 1 mM EDTA, and 1 mM EGTA) supplemented with 1 mM PMSF, 1 mM β-glycerophosphate, and 1X protein inhibitor cocktail (PIC). Lysates were clarified by centrifugation at maximum speed. The protein concentration of supernatant was determined using Bradford assay. After 500 μg of proteins in IP lysis buffer were rotated overnight at 4 °C with α-Myc antibody, they were then rotated with 20 μl of 50% slurry of protein A/G PLUS-Agarose (Santa Cruz Biotechnology) for 3 h at 4 °C. Immunoprecipitated proteins were collected by centrifugation and then washed three times with lysis buffer. These beads were then mixed with SDS sample buffer and prepared for SDS-PAGE.

### Western blot-SUnSET analysis

Ssu72 WT and aKO mice were injected with puromycin dihydrochloride (Sigma-Aldrich, #P9620) at a dose of 40 nmol/g of body weight and then sacrificed at 30 min after administration[47]. BAT tissue lysates from mice were prepared for SDS-PAGE analysis. Western blotting was conducted as described above. The membrane was blocked with 5% skim milk. To monitor mRNA translation, the membrane was incubated with primary antibody (DSHB, #PMY-2A4) diluted to 1:200 in 5% BSA for detecting puromycin-labeled peptides. These membranes were washed with TBS-T and incubated with anti-mouse IgG-HRP secondary antibody (GenDEPOT, #SA001-500) diluted to 1:7000 in 5% skim milk.

### Additional resource

Analyzed cell type-enriched transcripts datasets with expression profiles[83] were obtained from Human Protein Atlas website (https://www.proteinatlas.org/humanproteome/tissue+cell+type). Differential correlation score between each gene and the cell-type-enriched transcripts was analyzed.

### Statistical analysis

Results were analyzed using two-tailed Student's $t$ test, one-way ANOVA, or two-way ANOVA where appropriate using GraphPad Prism software (version 7). A Bonferroni post hoc test was used to test for significant differences as determined by ANOVA. Significance was accepted at $p < 0.05$. Data are presented as mean ± SD or SEM.

### Reporting summary

Further information on research design is available in the Nature Portfolio Reporting Summary linked to this article.

## Data availability

The human transcriptomic datasets used in this study are available in the GEO database under accession code GSE2508. The raw RNA-seq data generated in this study have been deposited in the GEO database under accession code GSE224337. List of differentially expressed genes (DEGs) in BAT between Ssu72 WT and aKO mice from RNA-seq data is provided in Supplementary Data 1. The molecular signature databases (MSigDB) used for GSEA are available on the GSEA MSigDB website [https://www.gsea-msigdb.org/gsea/msigdb/index.jsp]. Raw data for all figures are provided in the Source Data file. Source data are provided with this paper.

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

## Acknowledgements

This study was supported by grants (NRF-2022R1A2B5B03001431 and NRF-2022M3A9H1014129 to C.-W.L.) of the National Research Foundation (NRF) funded by the Ministry of Education, Science, and Technology (MEST), Republic of Korea. M.-W.L. was supported by grant (NRF-2021R1A2C1005434) of NRF funded by MEST, Republic of Korea.

## Author contributions

E.-J.P. and H.-S.K. designed the studies, analyzed data, and wrote the manuscript. D.-H.L., S.-M.K., J.-S.Y., and J.-M.L. contributed to specific experiments. S.J.I. and H.L. provided materials and participated in data generation. H.-S.K., M.-W.L., and C.-W.L. designed the studies, supervised the overall project, wrote the manuscript, and performed the final manuscript preparation. All authors provided feedback and agreed on the final manuscript.

## Competing interests

The authors declare no competing interests.
