## [Peer Review File · Nature Communications]

Ssu72 phosphatase is essential for thermogenic adaptation by regulating cytosolic translationREVIEWER COMMENTS

Reviewer #1 (Remarks to the Author):

Ying and colleagues have investigated the role of Ssu72 phosphatase in brown adipose tissue, showing its importance on translational control of mitochondrial function and brown fat mediated thermogenesis. By using a sophisticated set of methods, including adipose tissue specific Ssu72 knockout mice as well as aKOs with ectopic expression of Ssu72 as a rescue, the authors demonstrate the impact of Ssu72 on phosphorylation of eIF2 α , translation of OXPHOS subunits, mitochondrial morphology and finally on cold response.

This study provides convincing and clear data on the role of Ssu72 on BAT function. The data are of interest and mainly support the conclusions. However, a few issues and concerns that need to be addressed which are detailed below.

1. I have the feeling, starting the manuscript with a dataset of lean and obese humans is a little bit out of context. I agree that activation of thermogenic adipose tissues is associated with improved systemic metabolism, but to my knowledge, current data suggest that without significant expansion, BAT activation is insufficient to induce weight loss in humans. BAT activation may be a treatment for the comorbidities associated with obesity, such as hyperglycemia and dyslipidemia, rather than obesity itself. I understand that associations with human data are important, but wouldn't it be more fitting for the manuscript to look at cold vs warm (BAT) or WAT vs BAT human datasets instead obese vs lean?

2. aKO mice show already a very dramatic "whiting of BAT" under RT (cold stress for mice), but no differences in energy expenditure, body temperature as well as gene and protein expression of thermogenic markers (2c, 6b) or fatty acid oxidation (2e). However, the stress response as well as FGF21 are strongly increased. So, I was wondering if FGF21 levels in serum are increased, similar to the UCP1 KO mouse model (PMID: 28768181, PMID: 26137441). Furthermore, it is very interesting that, in contrast to UCP1 KO mice, no morphological signs of browning in white adipose tissue of aKO mice was detected. The question arises, what heat source do the animals use during RT. Are alternative thermogenic mechanisms induced? Or are there any signs of browning in WAT besides morphology.

3. Does the fact that WT and aKO mice have the same energy expenditure and body temperature at RT indicate that the animals can adapt to colder temperatures over time? (like UCP1 KO mice). To answer this, it would be interesting to perform a cold tolerance test from thermoneutral conditions to RT or colder. Or adapt animals slowly to the cold (23, 18, 10, 4°C).

4. There is already some literature on the importance of phosphatases and eIF2 α /ER stress on BAT function (f.e. PMID: 22509299, PMID: 27077115, PMID: 32029570, PMID: 30814564) with sometimes opposite results. Relevant ones could be discussed.

5. While data on Ssu72 and eIF2 α interaction are quite convincing the role of Ssu72 on transcriptional control of gene expression upon cold is less clear. Especially with regard to the summary cartoon, this should be discussed more in detail.

6. The authors conclude that Ssu72 is associated with mitochondrial translation. Did the overexpression of Ssu72 also rescue the OXPHOS protein expression as well as the cold tolerant phenotype? This would be essential for most conclusions. The video of one animal's cold activity per genotype is a clue, but nothing more. If this is just an n of 1, I would suggest to remove the figure (6e) and temper the conclusion on behaviour during cold.

7. Dephosphorylation of eIF2 α is central to ISR signal termination to restore protein synthesis and normal cell functioning, and is therefore tightly regulated. Is there any compensation by other phosphatases at RT in aKO mice? And what about feedback control mechanisms such as GADD34

or CReP?

Minor comments:

- Please show data of body weight development of Ssu72 floxed AdipoCre mice.
- line 145 fig 1g instead of 1f
- please do not normalize VO₂ to kg bodyweight (see PMID: 34489606, PMID: 22205519).
- Please show single values in the bar graphs (n number not always noted)

Reviewer #2 (Remarks to the Author):

This study shows that Ssu72 expression is up-regulated in BAT after cold exposure. Knockout of Ssu72 in BAT has an effect on cold-tolerance of mice and mitochondrial structure. Ssu72 also affects global translation through dephosphorylation of eIF2 α . There are many overstated conclusions throughout the manuscript. For example, in the abstract, the authors stated that Ssu72 phosphatase is a novel protein phosphatase, but Ssu72 has long been proven to be a phosphatase of PNAPII. In Page 5, the authors claimed that Ssu72 is specifically involved in translational control for mitochondrial function and thermogenesis in brown adipose tissues (Line 99-101). The specificity, however, has not been proven. Ssu72 affects global protein translation instead of a subset of specific proteins. Ssu72 is also expressed in many tissues, and in iWAT, Ssu72 expression levels also responded to cold exposure (Extended Data Fig. 1d).....

It should be noted that interfering with any important cellular process usually has an effect on mitochondria and consequently has a physiological phenotype. Interfering with global translation in a specific tissue is for sure going to disrupt mitochondrial functions and obstruct the normal function of the tissue. The only solid and novel conclusion of the manuscript is that Ssu72 is a phosphatase of eIF2 α .

In addition to being the phosphatase of eIF2 α , Ssu72 is also the phosphatase of PNAPII and probably many more, the authors haven't taken the effects of these other substrates and the corresponding pathways into consideration in their explanation of the knockout phenotypes.

There are also some additional concerns:

1. Fig. 1b, Ssu72 is not expressed only in BAT. Hearts probably have higher expression levels. How specific this response to temperature in BAT is hasn't been looked at.
2. Fig. 1d, GAPDH is a metabolic protein. Is it a good control in this case? From the figures shown, GAPDH levels at RT and cold are clearly not the same.
3. Fig. 2d, again, why use the heat shock protein as a control if the authors were studying cold response?
4. Fig. 4d, are the two lanes two animals? Clearly HSP90 levels are very different in the two WT samples.
5. Fig. 4f, the myc control also has some clear HA signal. Myc-Ssu72 IP controls are not equal either.
6. Fig. 6b, the quantification does not match the blots, especially for AMPK α .
7. Fig. 6c, and d, the Ssu72 blots do not show a three-fold increase of aKO;cTg compared to WT as shown in quantification graph.
8. Fig. 6, 7 and extended data Fig. 9, most of these results only have RT data. No cold exposure data were shown. If there was a three-fold increase of Ssu72 in aKO;cTg as claimed, which is probably

more relevant to the increase of Ssu72 levels in BAT after cold exposure, shouldn't the levels of the examined proteins be higher in aKO;cTg compared to WT in extended data Fig 9a?

Reviewer #1 (Remarks to the Author):

Ying and colleagues have investigated the role of Ssu72 phosphatase in brown adipose tissue, showing its importance on translational control of mitochondrial function and brown fat mediated thermogenesis. By using a sophisticated set of methods, including adipose tissue specific Ssu72 knockout mice as well as aKOs with ectopic expression of Ssu72 as a rescue, the authors demonstrate the impact of Ssu72 on phosphorylation of eIF2 α , translation of OXPHOS subunits, mitochondrial morphology and finally on cold response.

This study provides convincing and clear data on the role of Ssu72 on BAT function. The data are of interest and mainly support the conclusions. However, a few issues and concerns that need to be addressed which are detailed below.

Response: We thank this reviewer for valuable comments, which has helped us strengthen our conclusions. We have performed additional experiments and incorporated them into the revised manuscript.

1. I have the feeling, starting the manuscript with a dataset of lean and obese humans is a little bit out of context. I agree that activation of thermogenic adipose tissues is associated with improved systemic metabolism, but to my knowledge, current data suggest that without significant expansion, BAT activation is insufficient to induce weight loss in humans. BAT activation may be a treatment for the comorbidities associated with obesity, such as hyperglycemia and dyslipidemia, rather than obesity itself. I understand that associations with human data are important, but wouldn't it be more fitting for the manuscript to look at cold vs warm (BAT) or WAT vs BAT human datasets instead obese vs lean?

Response: We agree with the reviewer that human BAT dataset is more contextual in this study. Since the involvement of Ssu72 in metabolism, especially in adipose tissue metabolism, has not been elucidated yet, we have analyzed human public datasets to explore the significance of Ssu72 expression in adipocytes of human patients with metabolic diseases. We have also tried to find information on Ssu72 expression in human BAT datasets. However, we were unable to obtain information on Ssu72 expression in human BAT from public human BAT datasets under appropriate experimental conditions. Although recent studies have applied refined ¹⁸F-DG-PET-CT scan analysis to understand metabolic contribution of BAT in humans (van Marken Lichtenbelt, W. D. et al. *N Engl J Med* 2009, Leitner, B. P. et al. *Proc Natl Acad Sci U S A* 2017), there were limitations in distinguishing between thermogenic and non-thermogenic cells in human BAT (Perdikari, A. et al. *Cell Rep* 2018). Because adult human BAT depots are highly heterogeneous, different from mouse BAT depots (Shinoda, K. et al. *Nat Med* 2015), analyzing cell-specific gene expression using biopsied adult human BAT is highly difficult (de Jong, J. M. et al. *Am J Physiol Endocrinol Metab* 2015). In response to reviewer's comment, we have reorganized our data as shown in a supplementary figure (revised **Supplementary Fig. 1a**) and instead highlighted results of the cold-responsive Ssu72 in mice exposed to both mild and severe cold as main figure (revised **Fig. 1c, d**).

2. aKO mice show already a very dramatic "whiting of BAT" under RT (cold stress for mice), but no differences in energy expenditure, body temperature as well as gene and protein expression of thermogenic markers (2c, 6b) or fatty acid oxidation (2e). However, the stress response as well as FGF21 are strongly increased. So, I was wondering if FGF21 levels in

serum are increased, similar to the UCP1 KO mouse model (PMID: 28768181, PMID: 26137441). Furthermore, it is very interesting that, in contrast to UCP1 KO mice, no morphological signs of browning in white adipose tissue of aKO mice was detected. The question arises, what heat source do the animals use during RT. Are alternative thermogenic mechanisms induced? Or are there any signs of browning in WAT besides morphology.

Response: We thank the reviewer for these insightful questions. Considering the phenotype of Ssu72 aKO mouse, which still expresses UCP1 protein at low levels, some differences from UCP1 KO mouse may occur. To address the first question raised by the reviewer, we have evaluated FGF21 protein levels in Ssu72 aKO mice. As shown in **Fig. R1a** and **R1b** below, we found that Ssu72 aKO mice exhibited increased FGF21 protein levels in both BAT and serum compared to WT mice. However, these levels were relatively low compared to those in UCP1 KO mice (Keipert, S. et al. *Cell Metab* 2017, Keipert, S. et al. *Mol Metab* 2015), and the increase in the expression at such degree of difference was considered to be insufficient to induce WAT thermogenesis.

Not shown in the manuscript

Fig. R1 (Not shown in the manuscript): a, Protein expression levels of FGF21 and GAPDH in BAT from Ssu72 WT and aKO mice housed at room temperature (RT, 23°C) (n = 3 mice per group). b, FGF21 serum levels in WT (n = 5) and aKO mice (n = 8) housed at RT. Data are presented as mean \pm SEM. *p < 0.05. Statistical comparisons were made using Student's t-test.

Interestingly, in the paper cited by the reviewer, the authors demonstrated that FGF21 has minor effects on WAT remodeling and metabolic homeostasis during chronic cold acclimation (Keipert, S. et al. *Cell Metab* 2017). Additionally, we found that the expression of UCP1 protein in iWAT of WT mice decreased after 7-8 weeks of age (shown below, **Fig. R2a**). Thus, in the case of mild chronic cold adaptation (similar to housing mice at RT), the thermogenic effect by FGF21 might be less. Collectively, unlike the UCP1 KO mouse phenotype, the reason why Ssu72 aKO mice did not show WAT browning might be due to the cold adaptation. The question then arises as to what heat source Ssu72 aKO mice use under RT conditions, as suggested by the reviewer. Unlike UCP1 KO mice, Ssu72 aKO mice express some UCP1 proteins in BAT. In addition to this, we can consider UCP1-independent thermogenesis in adipose tissues. Notably, creatine energetics is one of the key effectors of UCP-1 independent adaptive thermogenesis (Kazak, L. et al. *Cell Metab* 2017, Chouchani, E. T. et al. *Cell Metab* 2019). Interestingly, iWAT RNA-seq results from Ssu72 WT and aKO mice revealed greatly increased expression of two creatine kinase isoforms, *Ckmt2* and *Ckm*, in Ssu72 aKO mice compared to WT mice (shown below, **Fig. R2b**). This result suggests that Ssu72 aKO mice might be able to maintain their body temperature by increasing futile creatine cycling-based thermogenesis in iWAT under RT conditions.

Not shown in the manuscript

Fig. R2 (Not shown in the manuscript): a, Protein expression levels of HSP90 (loading control) and UCP1 in iWAT from WT mice housed at room temperature (RT, 23°C) (n = 4 mice per group). b, Relative fold changes expressed as log₂ of normalized average count (RNA-seq) of each gene in aKO iWAT samples divided by that in WT iWAT (n = 3 mice per group).

3. Does the fact that WT and aKO mice have the same energy expenditure and body temperature at RT indicate that the animals can adapt to colder temperatures over time? (like UCP1 KO mice). To answer this, it would be interesting to perform a cold tolerance test from thermoneutral conditions to RT or colder. Or adapt animals slowly to the cold (23, 18, 10, 4°C).

Response: We appreciate the reviewer's suggestion. To address this question, we have measured body temperatures of Ssu72 WT and aKO mice during a slow decrease in ambient temperature from thermoneutral to severe cold conditions (30, 23, 18, 10, 5°C), as shown in **Fig. R3**.

Fig. R3: Schematic illustration of cold acclimation (from 30°C to 5°C). Mice were housed at 30°C for 10 days, followed by cold exposure at 23, 18, 10, 5°C for 2 days.

As a result, the average body temperature of Ssu72 WT mice was approximately 36.7°C at the beginning of measurement and decreased to 35.5°C after chronic cold adaptation. However, in Ssu72 aKO mice, body temperature was maintained during chronic cold adaptation as shown in **Fig. R4a**. We hypothesized that the reason for the relatively high body temperature in Ssu72 aKO mice during chronic cold adaptation could be due to increased futile creatine metabolism in iWAT, as mentioned above. Indeed, we found that mRNA expression level of creatine kinase *Ckmt2* was higher in Ssu72 aKO mice upon cold adaptation than in WT mice as shown in **Fig. R4b**. Therefore, the result of body temperature in Ssu72 aKO mice during chronic cold adaptation may be due to thermogenesis by slightly expressed UCP1 as well as increased futile creatine cycling-based thermogenesis.

Not shown in the manuscript

Fig. R4 (Not shown in the manuscript): Ssu72 WT and Ssu72 aKO mice exposed to chronic cold exposure (from 30°C to 5°C) (n = 3-4 mice). **a**, Body temperature of mice during the first 6 hours of each temperature after exposure to cold. Data are presented as mean \pm SEM. * $p < 0.05$. Statistical comparisons were made using two-way ANOVA. **b**, Relative mRNA expression levels of *Ssu72*, *Ucp1*, and *Ckmt2* in iWAT from mice exposed to chronic cold (from 30°C to 5°C). Data are presented as mean \pm SEM. * $p < 0.05$, *** $p < 0.001$. Statistical comparisons were made using Student's t-test.

4. There is already some literature on the importance of phosphatases and eIF2 α /ER stress on BAT function (f.e. PMID: 22509299, PMID: 27077115, PMID: 32029570, PMID: 30814564) with sometimes opposite results. Relevant ones could be discussed.

Response: We thank the reviewer for this great suggestion, and we agree with the reviewer. Therefore, we have incorporated it in the revised manuscript. Previous studies have revealed the importance of several eIF2 α phosphatases for energy homeostasis. GADD34, a well-known eIF2 α phosphatase, has been shown to have complex effects on resistance to diet-induced

obesity (Oyadomari, S. et al. *Cell Metab* 2008, Nishio, N. et al. *Sci Rep* 2015, Patel, V. et al. *Sci Rep* 2019). Notably, the paper by Patel et al. (*Sci Rep* 2019, PMID: 30814564) showed that loss of GADD34 decreased the consumption of high-fat food, consistent with an effect of hypothalamic eIF2 α phosphorylation in regulating feeding behaviors (Maurin, A. et al. *Cell Rep* 2014). Furthermore, the paper by Bettaieb et al. (*PLoS One* 2012, PMID: 22509299) demonstrated that PTP1B dephosphorylates PERK and regulates ER stress in brown adipocytes. Although this paper showed that PTP1B deficiency could attenuate protein synthesis in brown adipocytes, how translational regulation through eIF2 α phosphorylation can directly affect metabolic adaptation of brown adipocytes remains unclear. In addition, it has been revealed that PERK can coordinate mitochondrial molecular quality control in response to ER stress (Lebeau, J. et al. *Cell Rep* 2018). Under ER and nutrient stress condition, PERK-eIF2 α pathway can promote respiratory supercomplex assembly through SCAF1. However, this pathway is not sufficient to form cristae (Balsa, E. et al. *Mol Cell* 2019). The paper by Kato et al. (*Life Sci Alliance* 2020, PMID: 32029570) showed that PERK was phosphorylated during brown adipocyte differentiation, contributing to mitochondrial cristae formation through PERK-GABP α pathway during differentiation. Consistent with these findings, Latorre-Muro et al. (*Cell Metab* 2021) has demonstrated that PERK promotes cristae formation through MIC19 protein import and that induction of MIC19 is independent of the canonical PERK-eIF2 α pathway. We have added these citations to the appropriate paragraphs in the revised manuscript (**Introduction, line 73-82; Discussion, line 483-493**).

One of the papers cited by the reviewer (Sekine, S. et al. *EBioMedicine* 2016. PMID: 27077115) showed the phenotype of mitochondrial protein PGAM5 whole-body knockout mouse. It is noteworthy that the experimental condition used in that study is subjecting mice to cold challenge after 12 hours of fasting. Since lipolysis of adipose tissue has already proceeded during fasting, it is a limited condition in which substrates required for BAT function are insufficient. For this reason, WT mice showed vulnerability during cold exposure after fasting. Furthermore, a recent study has demonstrated that PGAM5 can repress UCP1 expression and activity (Sugawara, S. et al. *J Biol Chem* 2020). Considering the physiological function of PGAM5, it is thought that PGAM5 KO mice already had increased UCP1 expression (or iWAT browning) compared to WT mice, which might have allowed them to maintain their core body temperature and survive longer than WT mice during cold exposure after fasting. Additionally, it has been revealed that PGAM5 is required for mitochondrial fission and regulation of mitochondrial dynamics (Yu, B. et al. *Nat Commun* 2020). Therefore, the increase in P-eIF2 α signaling pathway in cold-exposed PGAM5 KO mice after fasting might be a result of mitochondrial dysfunction under excessive metabolic challenge with limited BAT function.

5. While data on Ssu72 and eIF2 α interaction are quite convincing the role of Ssu72 on transcriptional control of gene expression upon cold is less clear. Especially with regard to the summary cartoon, this should be discussed more in detail.

Response: We thank the reviewer for this comment and apologize for our unclear explanation. Transcriptional control of thermogenic genes upon severe cold exposure shown in Ssu72-deficient BAT is thought to be a secondary effect, indirectly due to mitochondrial dysfunction in BAT. BAT mitochondria can communicate with nucleus and affect transcription of genes depending on the status of respiratory capacity (Yun, J. et al. *Cell Metab* 2014, Nam, M. et al. *Front Endocrinol* 2015). Several studies have shown that impaired mitochondrial respiratory capacity results in decreased expression of thermogenic genes in BAT (Nam, M. et al. *Sci Rep* 2017, Ikeda, K. *Nat Commun* 2013, Cooper, M. P. et al. *J Biol Chem* 2008), and low respiratory

activity may lead to lipid accumulation by affecting fatty acid oxidation (Liu, L. et al. *J Biol Chem* 2011). Interestingly, it was observed that a decrease in expression of mitochondrial OXPHOS protein was accompanied by upregulation of lipid biosynthetic genes in BAT of Ssu72 aKO mice compared to WT mice housed at RT (revised **Fig. 7b-g** and **Supplementary Fig. 6b**). These results suggest that the mitochondrial respiratory capacity in Ssu72-deficient brown adipocytes might be lower than that in WT brown adipocytes. Additionally, severe cold stimulation requires high respiration capacity in brown adipocytes and enhances thermogenic gene program (Chouchani, E. T. et al. *Cell Metab* 2019). Thus, in response to severe cold, expression of thermogenic genes was increased within a few hours in BAT of WT mice with normal respiratory capacity. In Ssu72 aKO mice, however, it is thought that thermogenic gene expression was not immediately increased in response to severe cold due to a secondary effect of reduced respiratory capacity. We have included this detailed explanation in the revised **Discussion (line 451-461)**.

6. *The authors conclude that Ssu72 is associated with mitochondrial translation. Did the overexpression of Ssu72 also rescue the OXPHOS protein expression as well as the cold tolerant phenotype? This would be essential for most conclusions. The video of one animal's cold activity per genotype is a clue, but nothing more. If this is just an n of 1, I would suggest to remove the figure (6e) and temper the conclusion on behaviour during cold.*

Response: Thanks for this comment and we agree with the reviewer. To address this question, we have assessed OXPHOS protein expression in BAT of Ssu72 WT, aKO, and aKO;cTg mice. We found that expression levels of OXPHOS complexes proteins encoded by both nuclear and mitochondrial DNA were rescued by ectopic Ssu72 expression in BAT of aKO;cTg mice (revised **Fig. 7f, g**). This result suggests that Ssu72 is important for OXPHOS protein expression. In addition, we agree with the reviewer that the short video shows a simple clue. Therefore, we have removed the graph about the distance travelled of mice during cold exposure. Although only one mouse per group is shown in the Supplementary Video, most mice exhibited similar behavior by genotype during acute cold exposure. To demonstrate differences in cold tolerance of mice, we have included the result of mouse survival rate during acute cold exposure (4°C) for 8 hours (revised **Supplementary Fig. 10c**). Ssu72 aKO mice became hypothermic within a few hours of cold exposure (**Fig. 2a**), and this severe cold susceptibility was lethal to aKO mice. However, ectopic Ssu72 expression in adipocytes sufficiently rescued the cold tolerance of mice (revised **Supplementary Fig. 10c**).

7. *Dephosphorylation of eIF2 α is central to ISR signal termination to restore protein synthesis and normal cell functioning, and is therefore tightly regulated. Is there any compensation by other phosphatases at RT in aKO mice? And what about feedback control mechanisms such as GADD34 or CReP?*

Response: We thank the reviewer for raising this important point. To address this question, we have examined expression levels of other well-known eIF2 α phosphatases, including GADD34, CReP, and PP1, in BAT of Ssu72 WT and aKO mice housed at RT. In aKO mice, only *Ppp1r15a* (GADD34) mRNA levels were upregulated, and there were no differences in *Ppp1r15b* (CReP) or *Ppp1ca* (PP1) mRNA levels (revised **Supplementary Fig. 8a**). The increased expression level of *Ppp1r15a* in aKO mice was probably due to compensation for hyperphosphorylation of eIF2 α . We agree with the reviewer's comment that other eIF2 α phosphatases might act in

feedback control on hyperphosphorylation of eIF2 α in Ssu72 aKO mice. However, because eIF2 α phosphorylation was upregulated in BAT of aKO mice despite increased GADD34 expression, effects of the other eIF2 α phosphatases might be limited in BAT. To support this point, we next investigated whether gene expression levels of these phosphatases were different for each tissue cell type. Analyzed data of correlation score between each gene and cell-type-enriched transcripts were obtained from Human Protein Atlas (<https://www.proteinatlas.org/>). Notably, we found that only Ssu72 was predominantly enriched in adipocytes of both subcutaneous and visceral fat tissues, while other eIF2 α phosphatases were not (revised **Supplementary Fig. 8b**). This result suggests that the effect of Ssu72 phosphatase on eIF2 α dephosphorylation in adipocytes may be greater than those of other phosphatases.

Minor comments:

- *Please show data of body weight development of SSU72 flox AdipoCre mice.*

Response: We have included a new data on body weight gain of Ssu72 WT and aKO littermates (**Supplementary Fig. 2d**).

- *line 145 fig 1g instead of 1f*

Response: We have corrected this typo.

- *please do not normalize VO₂ to kg bodyweight (see PMID: 34489606, PMID: 22205519).*

Response: We thank the reviewer for this suggestion. We have changed the data to not divide oxygen consumption by body mass (**Fig. 2f**).

- *Please show single values in the bar graphs (n number not always noted)*

Response: We have replaced all bar graphs with plots and added information about the number of mice in the figure legend. The raw data underlying all graphs are provided in **Source Data**.

Reviewer #2 (Remarks to the Author):

This study shows that Ssu72 expression is up-regulated in BAT after cold exposure. Knockout of Ssu72 in BAT has an effect on cold-tolerance of mice and mitochondrial structure. Ssu72 also affects global translation through dephosphorylation of eIF2 α . There are many overstated conclusions throughout the manuscript. For example, in the abstract, the authors stated that Ssu72 phosphatase is a novel protein phosphatase, but Ssu72 has long been proven to be a phosphatase of RNAPII. In Page 5, the authors claimed that Ssu72 is specifically involved in translational control for mitochondrial function and thermogenesis in brown adipose tissues (Line 99-101). The specificity, however, has not been proven. Ssu72 affects global protein translation instead of a subset of specific proteins. Ssu72 is also expressed in many tissues, and in iWAT, Ssu72 expression levels also responded to cold exposure (Extended Data Fig. 1d).....

Response: We thank this reviewer for these detailed comments. We agree that Ssu72 is a well-known phosphatase of RNAPII, and we have revised the text to clarify that what we have uncovered throughout this paper is a novel function of Ssu72 phosphatase in BAT thermogenesis (**line 102-104**). Cold stimulation induces β_3 -adrenergic signaling, especially in thermogenic adipocytes (brown and beige adipocytes) that express β_3 -adrenergic receptor (Cypess, A. M. et al. *Cell Metab* 2015, Inagaki, T. et al. *Nat Rev Mol Cell Biol* 2016). In addition, when exposed to cold, brown adipocytes are activated within a short period of time via the sympathetic nervous system to rapidly adapt to changes in ambient temperature (Ouellet, V. et al. *J Clin Invest* 2012, Betz, M. J. et al. *Nat Rev Endocrinol* 2018). In this paper, we revealed that Ssu72 expression was enriched in BAT at RT and increased in response to acute cold. We also found that Ssu72 expression was also increased in iWAT of WT mice after 12 hours of cold exposure (**Supplementary Fig. 2d**). However, Ssu72 aKO mice exhibited a rapid loss of body temperature and fatal hypothermia within 6 hours of exposing to severe cold (**Fig. 2a** and revised **Supplementary Fig. 10c**). Therefore, cold susceptibility of Ssu72 aKO mice during acute cold exposure was likely affected by brown adipocyte dysfunction.

It should be noted that interfering with any important cellular process usually has an effect on mitochondria and consequently has a physiological phenotype. Interfering with global translation in a specific tissue is for sure going to disrupt mitochondrial functions and obstruct the normal function of the tissue. The only solid and novel conclusion of the manuscript is that Ssu72 is a phosphatase of eIF2 α .

Response: We have carefully considered the point raised by the reviewer. Regulation of eIF2 α phosphorylation by Ssu72 in BAT appears to be important for maintaining body temperature through translation of key effectors of thermogenesis. To address the reviewer's question, we have investigated effects of Ssu72 on mitochondrial function and thermogenesis in a non-adipose tissue such as liver, where mitochondrial function is important for metabolic health of the tissue. We used hepatocyte-specific Ssu72 knockout mouse model (Albumin-Cre; Ssu72^{flox/flox}) for cold challenge experiment. After Ssu72^{flox/flox} (Ssu72^{f/f}) and Albumin-Cre; Ssu72^{flox/flox} (Alb-Cre; Ssu72^{f/f}) mice were exposed to acute severe cold (4°C) for 6 hours, we found that rectal temperatures of Alb-Cre; Ssu72^{f/f} mice were similar to those of Ssu72^{f/f} mice during cold exposure (shown below, **Fig. R5a**). This result indicates that cold tolerance is not altered by hepatocyte-specific Ssu72 deletion. Considering the points raised by the reviewer, we additionally determined changes in phosphorylation status of eIF2 α and RNAPII CTD by Ssu72 deletion in the liver. Interestingly, as shown in **Fig. R5b**, there was no significant difference in the eIF2 α phosphorylation level by Ssu72 depletion in liver. In addition, we have

revealed that the effect Ssu72 on eIF2 α dephosphorylation, especially in adipocytes, might be greater than those of other eIF2 α phosphatases (revised **Supplementary Fig. 8b**). These results imply that the regulation of eIF2 α by Ssu72 phosphatase is more likely to be regulated in a tissue-specific manner rather than a global regulation. When comparing phosphorylation levels of both Ser5 and Ser7 of RNAPII CTD in hepatocytes, P-Ser5 showed a slight increase, but P-Ser7 did not show a significant difference (shown below, **Fig. R5b**). Thus, hepatic transcriptional regulation is not significantly affected by Ssu72.

Not shown in the manuscript

Fig. R5 (Not shown in the manuscript): **a**, Rectal core body temperatures of 7-8 weeks old Ssu72^{f/f} and Albumin-Cre; Ssu72^{f/f} (Alb-Cre; Ssu72^{f/f}) mice under cold conditions (4°C) (n = 4-5 mice per group). Data are presented as mean \pm SEM. Statistical comparisons were made using multiple t-tests. **b**, Western blots of lysates from liver of 8-week-old Ssu72^{f/f} and Alb-Cre; Ssu72^{f/f} littermates exposed to acute cold (4°C) for 6 hours.

We further investigated expression levels of mitochondrial OXPHOS complex subunits in liver of Ssu72^{f/f} and Alb-Cre; Ssu72^{f/f} mice. We found that there was no significant difference in

Not shown in the manuscript

Fig. R6 (Not shown in the manuscript): Western blots of OXPHOS complex subunits in liver of 8-week-old Ssu72^{f/f} and Alb-Cre; Ssu72^{f/f} littermates exposed to acute cold (4°C) for 6 hours.

protein expression of OXPHOS complex between the genotypes, as shown in **Fig. R6**. Together, these results demonstrate that cold-responsive Ssu72 is involved in the translation of key thermogenic effectors, particularly in BAT, thus affecting mitochondrial function and thermogenesis.

In addition to being the phosphatase of eIF2 α , Ssu72 is also the phosphatase of RNAPII and probably many more, the authors haven't taken the effects of these other substrates and the corresponding pathways into consideration in their explanation of the knockout phenotypes.

Response: We agree with the reviewer that we should have considered effects of other Ssu72 substrates on BAT function. To address this point, we have investigated changes in phosphorylation status of other Ssu72 substrates such as RNAPII in BAT of Ss72 WT and aKO mice. While only P-Ser7 was slightly increased in Ssu72 aKO mice under basal conditions, RNAPII CTD phosphorylation (P-Ser5, P-Ser7) under cold conditions was not significantly different between WT and aKO mice (shown below, **Fig. R7a, b**), indicating that RNAPII transcriptional regulation by Ssu72 is not significantly affected in BAT. Additionally, protein expression of HNF4a, previously identified as a Ssu72 substrate (Kim, H. et al. *Cell Death Differ* 2022), was too low in BAT to compare the expression levels between genotypes (shown below, **Fig. R7c**).

Fig. R7 (Not shown in the manuscript): **a, b** Western blots in BAT of Ssu72 WT and aKO mice housed at RT (**a**) or exposed to acute cold (4°C) for 4 hours (**b**) (n = 3 mice per group). **c**, Western blots of HNF4 α in BAT from WT and aKO mice (n = 3) and liver from WT mice housed at RT. The liver of WT mice was used as positive control for HNF4 α expression.

We hypothesized that this result might be due to different localization of Ssu72 in brown adipocytes. In brown adipocytes, Ssu72 is predominantly localized in the cytoplasm (shown below, **Fig. R8a**). Different localization of Ssu72 in brown adipocytes was further confirmed by subcellular fractionation of BAT and liver of WT mice. In the liver, Ssu72 was detected in the fractionated extracts of both cytosol (C) and nucleus (N) by immunoblotting, and Ssu72

was more abundant in the cytosol (shown below, **Fig. R8b**). Interestingly, while some Ssu72 were detected in the nuclear fraction of liver lysates, most Ssu72 were detected in the cytosolic fraction in BAT (shown below, **Fig. R8b**). Therefore, in brown adipocytes, Ssu72 appears to play an important role in the cytosol. Collectively, novel findings of this study are that Ssu72 expression is increased in BAT in response to acute cold and that Ssu72 is involved in the translation of key thermogenic effectors such as OXPHOS complex by eIF2 α dephosphorylation in BAT, thereby maintaining mitochondrial function and thermogenesis during cold exposure.

Not shown in the manuscript

Fig. R8 (Not shown in the manuscript): a, Immunohistochemical analysis with α -Ssu72 in BATs of 12-week-old Ssu72 WT and aKO mice housed at RT. (BAT of Ssu72 aKO mice were used as negative control) **b**, Western blot of lysate from subcellular compartments (cytosol (C) and nucleus (N)) of BAT and liver of WT mice housed at RT.

There are also some additional concerns:

1. *Fig. 1b, Ssu72 is not expressed only in BAT. Hearts probably have higher expression levels. How specific this response to temperature in BAT is hasn't been looked at.*

Response: As mentioned above, cold exposure specifically induces β_3 -adrenergic signaling pathway in thermogenic adipose tissue. In addition, we found that expression levels of

mitochondrial OXPHOS complex subunits in other metabolic tissues, including heart, skeletal muscle, and liver, were not altered by cold exposure for 2 days (shown below, **Fig. R9**).

Not shown in the manuscript

Fig. R9 (Not shown in the manuscript): Western blots of mitochondrial OXPHOS complexes (CI-V) in heart, skeletal muscle, and liver of WT mice exposed to RT or Cold for 2 days.

BAT thermogenesis is essential for mammals to rapidly maintain body temperature during acute cold exposure. We showed that *Ssu72* expression is increased in BAT by acute cold exposure (revised **Fig. 1c,d**). To clarify the specific response of *Ssu72* to cold in BAT, we assessed changes in *Ssu72* expression in response to cold in other metabolic tissues. We found that expression levels of *Ssu72* in response to acute cold for 5 hours in other metabolic tissues (heart, skeletal muscle, and liver) were similar to those at RT (shown below, **Fig. R10**). These results indicate that *Ssu72* expression is upregulated upon cold exposure, especially in BAT.

Not shown in the manuscript

Fig. R10 (Not shown in the manuscript): WT mice housed at RT or exposed to cold (4°C) for 5 hours (n = 4 mice per group) **a**, Relative mRNA expression levels of *Ucp1* in BAT from WT mice housed at RT or exposed to cold. mean \pm SEM. ***p < 0.001 (Student's t-test). **b-d**, Western blots of *Ssu72* in heart (**b**), skeletal muscle (**c**), and liver (**d**) of WT mice housed at RT or exposed cold.

2. Fig. 1d, GAPDH is a metabolic protein. Is it a good control in this case? From the figures shown, GAPDH levels at RT and cold are clearly not the same.

Response: We agree with the reviewer that GAPDH is a metabolic protein. Indeed, GAPDH has been used as a Western blot loading control in BAT during acute cold exposure (Liu, D. et al. *J Clin Invest* 2016, Chitraju, C. et al. *Cell Rep* 2020) and chronic cold exposure (Shin, H. et al. *Cell Metab* 2017, Yau, W. W. et al. *iScience* 2021). To clarify the suitability of GAPDH as a loading control in BAT during cold exposure, we have assessed expression levels of other loading controls (β -actin and α -tubulin) in the same sample for comparison. β -actin and α -tubulin have also been used as loading controls in BAT during acute cold exposure (Albert, V. et al. *EMBO Mol Med* 2016, Han, J. S. et al. *Nat Commun* 2022). Considering the difference in expression level of each loading control under RT and cold conditions, we have further quantified relative Ssu72 protein expression levels normalized to each control. We found significant differences in Ssu72 expression between RT and cold conditions (Shown below, **Fig. R11**).

Relative to Fig. 1d

Fig. R11 (Relative to Fig. 1d): Western blots of lysates from BAT of C57BL/6J mice housed at RT (23°C) or exposed to acute cold (4°C) for 4 hours (left) (n = 3 mice per group), with quantification of Ssu72 levels (right). Graph shows quantification of Ssu72 protein levels normalized to GAPDH, β -actin, or α -tubulin protein level using ImageJ. Data are presented as mean \pm SEM. **p < 0.01, ***p < 0.001. Statistical comparisons were made using Student's t-test.

3. Fig. 2d, again, why use the heat shock protein as a control if the authors were studying cold response?

Response: We thank the reviewer for raising this important point. Some heat shock proteins are constitutively expressed and can be used as loading controls, but it is important to consider that certain conditions can alter protein expression. It is noteworthy that cold stimulation specifically induces β_3 -adrenergic signaling in adipocytes, and HSP90 protein expression is not altered by β_3 -adrenergic signaling in adipocytes during acute cold exposure (Gerhart-Hines, Z. et al. *Nature* 2013, Paulo, E. et al. *Sci Rep* 2018) or chronic cold exposure (Dijk, W. et al. *Elife* 2015, Kutuyavin, V. I. et al. *Proc Natl Acad Sci U S A* 2019). To clarify the suitability of HSP90 as a loading control in BAT, we have newly added expression of β -actin. We found a similar expression pattern between HSP90 and β -actin during acute cold exposure (**Fig. 2d**).

4. Fig. 4d, are the two lanes two animals? Clearly HSP90 levels are very different in the two WT samples.

Response: We thank the reviewer for this comment. The number of mice per group is two and each lane represents each mouse. We have also described information about the number of mice in each figure legend. To show clearer data, we have repeated the experiment considering the amount of loading control (HSP90) and replaced it with new data (**Fig. 4d**).

5. Fig. 4f, the myc control also has some clear HA signal. Myc-Ssu72 IP controls are not equal either.

Response: We apologize for this confusion. The reason why some HA signal appeared in the myc control under IP condition was probably because washing of immunoprecipitated proteins during IP assay was not sufficient. Therefore, we have repeated the Co-IP assay and replaced this figure with new data (**Fig. 4f**).

6. Fig. 6b, the quantification does not match the blots, especially for AMPK α .

Response: We have newly added results of other loading controls (β -actin and α -tubulin) and further quantified relative expression levels of thermogenic proteins (PGC-1 α , UCP1, AMPK α , PKA) normalized to the expression level of α -tubulin (**Fig. 6b**).

7. Fig. 6c, and d, the Ssu72 blots do not show a three-fold increase of aKO;cTg compared to WT as shown in quantification graph.

Response: We apologize for the confusion. Fig. 6c is the result of protein expression in BAT and Fig. 6d in the previous manuscript (shown in revised **Supplementary Fig. 10a**) is mRNA expression result, not the quantification graph of protein levels in Fig. 6c. Thus, although expression level of Ssu72 mRNA in aKO;cTg mice was increased compared to WT (revised **Supplementary Fig. 10a**), there was no significant difference in protein expression level of Ssu72 between WT and aKO;cTg mice in the same mouse groups (**Fig. 6c**). The reason why we presented mRNA expression levels of thermogenic genes in aKO;cTg mice was to show that transcription of thermogenic factors in aKO;cTg mice was not significantly different from that in WT mice under RT conditions (revised **Supplementary Fig. 10a**).

8. Fig. 6, 7 and extended data Fig. 9, most of these results only have RT data. No cold exposure data were shown. If there was a three-fold increase of Ssu72 in aKO;cTg as claimed, which is probably more relevant to the increase of Ssu72 levels in BAT after cold exposure, shouldn't the levels of the examined proteins be higher in aKO;cTg compared to WT in extended data Fig 9a?

Response: Ssu72 mRNA expression level was increased about 3-fold in aKO;cTg mice compared to WT mice (revised **Supplementary Fig. 10a**). However, this increase did not affect

Ssu72 protein expression in aKO;cTg mice, which showed a similar expression level to WT mice (**Fig. 6c**). The reason why we mainly showed RT data is that expression levels of some thermogenic proteins were already reduced in BAT of Ssu72 aKO mice under RT condition (**Fig. 6b**), and expression levels of OXPHOS complexes, which are essential enzymes for thermogenesis, were also decreased in BAT of Ssu72 aKO mice at RT (revised **Fig. 7c, e**). These results suggest that Ssu72 aKO mice are insufficient to induce thermogenesis in response to severe cold exposure. In aKO;cTg mice, COX4 proteins were not completely restored to the expression levels in WT mice (revised **Fig. 7f**). However, after 4 hours of cold exposure (4°C), the sluggish and shivering behavior of Ssu72 aKO mice was restored by ectopic Ssu72 expression in aKO;cTg mice (**Supplementary Video 1**). In addition, we assessed UCP1 expression in BAT during cold exposure and found that reduced protein expression of UCP1 in aKO mice was also restored in aKO;cTg mice under cold condition (revised **Fig. 6e**).

REVIEWERS' COMMENTS

Reviewer #1 (Remarks to the Author):

The authors have responded to all my comments and I have no further concerns.

Reviewer #2 (Remarks to the Author):

The authors have addressed most of my concerns. The new results have shown that cold exposure has no effect on Ssu72 expression in all other tissues but BAT and iWAT and that knockdown of Ssu72 has no effect in all the substrates in any examined tissues except in BAT, and no effect in any other substrates in BAT except eIF2 α . This kind of specificity is very interesting. The authors did attempt to find an explanation. However, Fig. R2B does not show the conclusion as stated in the response because the nuclear controls are not even. Even if there were minor subcellular Ssu72 localization differences between the two tissues, it still does not explain the big function differences of the major cytosolic pools in the two tissues. Personally, I don't think this question has to be addressed for the paper if the other conclusions are solid. I would like to see, however, how UCP1 levels are changed in all the other tissues, since UCP1 levels have been the major indicating factor throughout the manuscript. In addition, the authors have done many experiments to address the concerns, but left the results out of the manuscript. I think it would be nice to present a more complete set of data and let the readers draw their own conclusion, instead of having the selected set and trying to convince readers of a conclusion that was actually conditional. This, however, is for the editors to decide.

Reviewer #1 (Remarks to the Author):

Ying and colleagues have investigated the role of Ssu72 phosphatase in brown adipose tissue, showing its importance on translational control of mitochondrial function and brown fat mediated thermogenesis. By using a sophisticated set of methods, including adipose tissue specific Ssu72 knockout mice as well as aKOs with ectopic expression of Ssu72 as a rescue, the authors demonstrate the impact of Ssu72 on phosphorylation of eIF2 α , translation of OXPHOS subunits, mitochondrial morphology and finally on cold response.

This study provides convincing and clear data on the role of Ssu72 on BAT function. The data are of interest and mainly support the conclusions. However, a few issues and concerns that need to be addressed which are detailed below.

Response: We thank this reviewer for valuable comments, which has helped us strengthen our conclusions. We have performed additional experiments and incorporated them into the revised manuscript.

1. I have the feeling, starting the manuscript with a dataset of lean and obese humans is a little bit out of context. I agree that activation of thermogenic adipose tissues is associated with improved systemic metabolism, but to my knowledge, current data suggest that without significant expansion, BAT activation is insufficient to induce weight loss in humans. BAT activation may be a treatment for the comorbidities associated with obesity, such as hyperglycemia and dyslipidemia, rather than obesity itself. I understand that associations with human data are important, but wouldn't it be more fitting for the manuscript to look at cold vs warm (BAT) or WAT vs BAT human datasets instead obese vs lean?

Response: We agree with the reviewer that human BAT dataset is more contextual in this study. Since the involvement of Ssu72 in metabolism, especially in adipose tissue metabolism, has not been elucidated yet, we have analyzed human public datasets to explore the significance of Ssu72 expression in adipocytes of human patients with metabolic diseases. We have also tried to find information on Ssu72 expression in human BAT datasets. However, we were unable to obtain information on Ssu72 expression in human BAT from public human BAT datasets under appropriate experimental conditions. Although recent studies have applied refined ¹⁸F-DG-PET-CT scan analysis to understand metabolic contribution of BAT in humans (van Marken Lichtenbelt, W. D. et al. *N Engl J Med* 2009, Leitner, B. P. et al. *Proc Natl Acad Sci U S A* 2017), there were limitations in distinguishing between thermogenic and non-thermogenic cells in human BAT (Perdikari, A. et al. *Cell Rep* 2018). Because adult human BAT depots are highly heterogenous, different from mouse BAT depots (Shinoda, K. et al. *Nat Med* 2015), analyzing cell-specific gene expression using biopsied adult human BAT is highly difficult (de Jong, J. M. et al. *Am J Physiol Endocrinol Metab* 2015). In response to reviewer's comment, we have reorganized our data as shown in a supplementary figure (revised **Supplementary Fig. 1a**) and instead highlighted results of the cold-responsive Ssu72 in mice exposed to both mild and severe cold as main figure (revised **Fig. 1c, d**).

2. aKO mice show already a very dramatic "whiting of BAT" under RT (cold stress for mice), but no differences in energy expenditure, body temperature as well as gene and protein expression of thermogenic markers (2c, 6b) or fatty acid oxidation (2e). However, the stress

response as well as FGF21 are strongly increased. So, I was wondering if FGF21 levels in serum are increased, similar to the UCP1 KO mouse model (PMID: 28768181, PMID: 26137441). Furthermore, it is very interesting that, in contrast to UCP1 KO mice, no morphological signs of browning in white adipose tissue of aKO mice was detected. The question arises, what heat source do the animals use during RT. Are alternative thermogenic mechanisms induced? Or are there any signs of browning in WAT besides morphology.

Response: We thank the reviewer for these insightful questions. Considering the phenotype of Ssu72 aKO mouse, which still expresses UCP1 protein at low levels, some differences from UCP1 KO mouse may occur. To address the first question raised by the reviewer, we have evaluated FGF21 protein levels in Ssu72 aKO mice. As shown in **Fig. R1a** and **R1b** below, we found that Ssu72 aKO mice exhibited increased FGF21 protein levels in both BAT and serum compared to WT mice. However, these levels were relatively low compared to those in UCP1 KO mice (Keipert, S. et al. *Cell Metab* 2017, Keipert, S. et al. *Mol Metab* 2015), and the increase in the expression at such degree of difference was considered to be insufficient to induce WAT thermogenesis.

Not shown in the manuscript

Fig. R1 (Not shown in the manuscript): a, Protein expression levels of FGF21 and GAPDH in BAT from Ssu72 WT and aKO mice housed at room temperature (RT, 23°C) (n = 3 mice per group). b, FGF21 serum levels in WT (n = 5) and aKO mice (n = 8) housed at RT. Data are presented as mean \pm SEM. *p < 0.05. Statistical comparisons were made using Student's t-test.

Interestingly, in the paper cited by the reviewer, the authors demonstrated that FGF21 has minor effects on WAT remodeling and metabolic homeostasis during chronic cold acclimation (Keipert, S. et al. *Cell Metab* 2017). Additionally, we found that the expression of UCP1 protein in iWAT of WT mice decreased after 7-8 weeks of age (shown below, **Fig. R2a**). Thus, in the case of mild chronic cold adaptation (similar to housing mice at RT), the thermogenic effect by FGF21 might be less. Collectively, unlike the UCP1 KO mouse phenotype, the reason why Ssu72 aKO mice did not show WAT browning might be due to the cold adaptation. The question then arises as to what heat source Ssu72 aKO mice use under RT conditions, as suggested by the reviewer. Unlike UCP1 KO mice, Ssu72 aKO mice express some UCP1 proteins in BAT. In addition to this, we can consider UCP1-independent thermogenesis in adipose tissues. Notably, creatine energetics is one of the key effectors of UCP-1 independent adaptive thermogenesis (Kazak, L. et al. *Cell Metab* 2017, Chouchani, E. T. et al. *Cell Metab* 2019). Interestingly, iWAT RNA-seq results from Ssu72 WT and aKO mice revealed greatly increased expression of two creatine kinase isoforms, *Ckmt2* and *Ckm*, in Ssu72 aKO mice compared to WT mice (shown below, **Fig. R2b**). This result suggests that Ssu72 aKO mice might be able to maintain their body temperature by increasing futile creatine cycling-based thermogenesis in iWAT under RT conditions.

Not shown in the manuscript

Fig. R2 (Not shown in the manuscript): a, Protein expression levels of HSP90 (loading control) and UCP1 in iWAT from WT mice housed at room temperature (RT, 23°C) (n = 4 mice per group). b, Relative fold changes expressed as log₂ of normalized average count (RNA-seq) of each gene in aKO iWAT samples divided by that in WT iWAT (n = 3 mice per group).

3. Does the fact that WT and aKO mice have the same energy expenditure and body temperature at RT indicate that the animals can adapt to colder temperatures over time? (like UCP1 KO mice). To answer this, it would be interesting to perform a cold tolerance test from thermoneutral conditions to RT or colder. Or adapt animals slowly to the cold (23, 18, 10, 4°C).

Response: We appreciate the reviewer's suggestion. To address this question, we have measured body temperatures of Ssu72 WT and aKO mice during a slow decrease in ambient temperature from thermoneutral to severe cold conditions (30, 23, 18, 10, 5°C), as shown in **Fig. R3**.

Fig. R3: Schematic illustration of cold acclimation (from 30°C to 5°C). Mice were housed at 30°C for 10 days, followed by cold exposure at 23, 18, 10, 5°C for 2 days.

As a result, the average body temperature of Ssu72 WT mice was approximately 36.7°C at the beginning of measurement and decreased to 35.5°C after chronic cold adaptation. However, in Ssu72 aKO mice, body temperature was maintained during chronic cold adaptation as shown in **Fig. R4a**. We hypothesized that the reason for the relatively high body temperature in Ssu72 aKO mice during chronic cold adaptation could be due to increased futile creatine metabolism in iWAT, as mentioned above. Indeed, we found that mRNA expression level of creatine kinase *Ckmt2* was higher in Ssu72 aKO mice upon cold adaptation than in WT mice as shown in **Fig. R4b**. Therefore, the result of body temperature in Ssu72 aKO mice during chronic cold adaptation may be due to thermogenesis by slightly expressed UCP1 as well as increased futile creatine cycling-based thermogenesis.

Not shown in the manuscript

Fig. R4 (Not shown in the manuscript): Ssu72 WT and Ssu72 aKO mice exposed to chronic cold exposure (from 30°C to 5°C) (n = 3-4 mice). **a**, Body temperature of mice during the first 6 hours of each temperature after exposure to cold. Data are presented as mean \pm SEM. * $p < 0.05$. Statistical comparisons were made using two-way ANOVA. **b**, Relative mRNA expression levels of *Ssu72*, *Ucp1*, and *Ckmt2* in iWAT from mice exposed to chronic cold (from 30°C to 5°C). Data are presented as mean \pm SEM. * $p < 0.05$, *** $p < 0.001$. Statistical comparisons were made using Student's t-test.

4. There is already some literature on the importance of phosphatases and eIF2 α /ER stress on BAT function (f.e. PMID: 22509299, PMID: 27077115, PMID: 32029570, PMID: 30814564) with sometimes opposite results. Relevant ones could be discussed.

Response: We thank the reviewer for this great suggestion, and we agree with the reviewer. Therefore, we have incorporated it in the revised manuscript. Previous studies have revealed the importance of several eIF2 α phosphatases for energy homeostasis. GADD34, a well-known eIF2 α phosphatase, has been shown to have complex effects on resistance to diet-induced

obesity (Oyadomari, S. et al. *Cell Metab* 2008, Nishio, N. et al. *Sci Rep* 2015, Patel, V. et al. *Sci Rep* 2019). Notably, the paper by Patel et al. (*Sci Rep* 2019, PMID: 30814564) showed that loss of GADD34 decreased the consumption of high-fat food, consistent with an effect of hypothalamic eIF2 α phosphorylation in regulating feeding behaviors (Maurin, A. et al. *Cell Rep* 2014). Furthermore, the paper by Bettaieb et al. (*PLoS One* 2012, PMID: 22509299) demonstrated that PTP1B dephosphorylates PERK and regulates ER stress in brown adipocytes. Although this paper showed that PTP1B deficiency could attenuate protein synthesis in brown adipocytes, how translational regulation through eIF2 α phosphorylation can directly affect metabolic adaptation of brown adipocytes remains unclear. In addition, it has been revealed that PERK can coordinate mitochondrial molecular quality control in response to ER stress (Lebeau, J. et al. *Cell Rep* 2018). Under ER and nutrient stress condition, PERK-eIF2 α pathway can promote respiratory supercomplex assembly through SCAF1. However, this pathway is not sufficient to form cristae (Balsa, E. et al. *Mol Cell* 2019). The paper by Kato et al. (*Life Sci Alliance* 2020, PMID: 32029570) showed that PERK was phosphorylated during brown adipocyte differentiation, contributing to mitochondrial cristae formation through PERK-GABP α pathway during differentiation. Consistent with these findings, Latorre-Muro et al. (*Cell Metab* 2021) has demonstrated that PERK promotes cristae formation through MIC19 protein import and that induction of MIC19 is independent of the canonical PERK-eIF2 α pathway. We have added these citations to the appropriate paragraphs in the revised manuscript (**Introduction, line 73-82; Discussion, line 483-493**).

One of the papers cited by the reviewer (Sekine, S. et al. *EBioMedicine* 2016. PMID: 27077115) showed the phenotype of mitochondrial protein PGAM5 whole-body knockout mouse. It is noteworthy that the experimental condition used in that study is subjecting mice to cold challenge after 12 hours of fasting. Since lipolysis of adipose tissue has already proceeded during fasting, it is a limited condition in which substrates required for BAT function are insufficient. For this reason, WT mice showed vulnerability during cold exposure after fasting. Furthermore, a recent study has demonstrated that PGAM5 can repress UCP1 expression and activity (Sugawara, S. et al. *J Biol Chem* 2020). Considering the physiological function of PGAM5, it is thought that PGAM5 KO mice already had increased UCP1 expression (or iWAT browning) compared to WT mice, which might have allowed them to maintain their core body temperature and survive longer than WT mice during cold exposure after fasting. Additionally, it has been revealed that PGAM5 is required for mitochondrial fission and regulation of mitochondrial dynamics (Yu, B. et al. *Nat Commun* 2020). Therefore, the increase in P-eIF2 α signaling pathway in cold-exposed PGAM5 KO mice after fasting might be a result of mitochondrial dysfunction under excessive metabolic challenge with limited BAT function.

5. While data on Ssu72 and eIF2 α interaction are quite convincing the role of Ssu72 on transcriptional control of gene expression upon cold is less clear. Especially with regard to the summary cartoon, this should be discussed more in detail.

Response: We thank the reviewer for this comment and apologize for our unclear explanation. Transcriptional control of thermogenic genes upon severe cold exposure shown in Ssu72-deficient BAT is thought to be a secondary effect, indirectly due to mitochondrial dysfunction in BAT. BAT mitochondria can communicate with nucleus and affect transcription of genes depending on the status of respiratory capacity (Yun, J. et al. *Cell Metab* 2014, Nam, M. et al. *Front Endocrinol* 2015). Several studies have shown that impaired mitochondrial respiratory capacity results in decreased expression of thermogenic genes in BAT (Nam, M. et al. *Sci Rep* 2017, Ikeda, K. *Nat Commun* 2013, Cooper, M. P. et al. *J Biol Chem* 2008), and low respiratory

activity may lead to lipid accumulation by affecting fatty acid oxidation (Liu, L. et al. *J Biol Chem* 2011). Interestingly, it was observed that a decrease in expression of mitochondrial OXPHOS protein was accompanied by upregulation of lipid biosynthetic genes in BAT of Ssu72 aKO mice compared to WT mice housed at RT (revised **Fig. 7b-g** and **Supplementary Fig. 6b**). These results suggest that the mitochondrial respiratory capacity in Ssu72-deficient brown adipocytes might be lower than that in WT brown adipocytes. Additionally, severe cold stimulation requires high respiration capacity in brown adipocytes and enhances thermogenic gene program (Chouchani, E. T. et al. *Cell Metab* 2019). Thus, in response to severe cold, expression of thermogenic genes was increased within a few hours in BAT of WT mice with normal respiratory capacity. In Ssu72 aKO mice, however, it is thought that thermogenic gene expression was not immediately increased in response to severe cold due to a secondary effect of reduced respiratory capacity. We have included this detailed explanation in the revised **Discussion (line 451-461)**.

6. *The authors conclude that Ssu72 is associated with mitochondrial translation. Did the overexpression of Ssu72 also rescue the OXPHOS protein expression as well as the cold tolerant phenotype? This would be essential for most conclusions. The video of one animal's cold activity per genotype is a clue, but nothing more. If this is just an n of 1, I would suggest to remove the figure (6e) and temper the conclusion on behaviour during cold.*

Response: Thanks for this comment and we agree with the reviewer. To address this question, we have assessed OXPHOS protein expression in BAT of Ssu72 WT, aKO, and aKO;cTg mice. We found that expression levels of OXPHOS complexes proteins encoded by both nuclear and mitochondrial DNA were rescued by ectopic Ssu72 expression in BAT of aKO;cTg mice (revised **Fig. 7f, g**). This result suggests that Ssu72 is important for OXPHOS protein expression. In addition, we agree with the reviewer that the short video shows a simple clue. Therefore, we have removed the graph about the distance travelled of mice during cold exposure. Although only one mouse per group is shown in the Supplementary Video, most mice exhibited similar behavior by genotype during acute cold exposure. To demonstrate differences in cold tolerance of mice, we have included the result of mouse survival rate during acute cold exposure (4°C) for 8 hours (revised **Supplementary Fig. 10c**). Ssu72 aKO mice became hypothermic within a few hours of cold exposure (**Fig. 2a**), and this severe cold susceptibility was lethal to aKO mice. However, ectopic Ssu72 expression in adipocytes sufficiently rescued the cold tolerance of mice (revised **Supplementary Fig. 10c**).

7. *Dephosphorylation of eIF2 α is central to ISR signal termination to restore protein synthesis and normal cell functioning, and is therefore tightly regulated. Is there any compensation by other phosphatases at RT in aKO mice? And what about feedback control mechanisms such as GADD34 or CReP?*

Response: We thank the reviewer for raising this important point. To address this question, we have examined expression levels of other well-known eIF2 α phosphatases, including GADD34, CReP, and PP1, in BAT of Ssu72 WT and aKO mice housed at RT. In aKO mice, only *Ppp1r15a* (GADD34) mRNA levels were upregulated, and there were no differences in *Ppp1r15b* (CReP) or *Ppp1ca* (PP1) mRNA levels (revised **Supplementary Fig. 8a**). The increased expression level of *Ppp1r15a* in aKO mice was probably due to compensation for hyperphosphorylation of eIF2 α . We agree with the reviewer's comment that other eIF2 α phosphatases might act in

feedback control on hyperphosphorylation of eIF2 α in Ssu72 aKO mice. However, because eIF2 α phosphorylation was upregulated in BAT of aKO mice despite increased GADD34 expression, effects of the other eIF2 α phosphatases might be limited in BAT. To support this point, we next investigated whether gene expression levels of these phosphatases were different for each tissue cell type. Analyzed data of correlation score between each gene and cell-type-enriched transcripts were obtained from Human Protein Atlas (<https://www.proteinatlas.org/>). Notably, we found that only Ssu72 was predominantly enriched in adipocytes of both subcutaneous and visceral fat tissues, while other eIF2 α phosphatases were not (revised **Supplementary Fig. 8b**). This result suggests that the effect of Ssu72 phosphatase on eIF2 α dephosphorylation in adipocytes may be greater than those of other phosphatases.

Minor comments:

- *Please show data of body weight development of SSU72 flox AdipoCre mice.*

Response: We have included a new data on body weight gain of Ssu72 WT and aKO littermates (**Supplementary Fig. 2d**).

- *line 145 fig 1g instead of 1f*

Response: We have corrected this typo.

- *please do not normalize VO2 to kg bodyweight (see PMID: 34489606, PMID: 22205519).*

Response: We thank the reviewer for this suggestion. We have changed the data to not divide oxygen consumption by body mass (**Fig. 2f**).

- *Please show single values in the bar graphs (n number not always noted)*

Response: We have replaced all bar graphs with plots and added information about the number of mice in the figure legend. The raw data underlying all graphs are provided in **Source Data**.

Reviewer #2 (Remarks to the Author):

This study shows that Ssu72 expression is up-regulated in BAT after cold exposure. Knockout of Ssu72 in BAT has an effect on cold-tolerance of mice and mitochondrial structure. Ssu72 also affects global translation through dephosphorylation of eIF2 α . There are many overstated conclusions throughout the manuscript. For example, in the abstract, the authors stated that Ssu72 phosphatase is a novel protein phosphatase, but Ssu72 has long been proven to be a phosphatase of RNAPII. In Page 5, the authors claimed that Ssu72 is specifically involved in translational control for mitochondrial function and thermogenesis in brown adipose tissues (Line 99-101). The specificity, however, has not been proven. Ssu72 affects global protein translation instead of a subset of specific proteins. Ssu72 is also expressed in many tissues, and in iWAT, Ssu72 expression levels also responded to cold exposure (Extended Data Fig. 1d).....

Response: We thank this reviewer for these detailed comments. We agree that Ssu72 is a well-known phosphatase of RNAPII, and we have revised the text to clarify that what we have uncovered throughout this paper is a novel function of Ssu72 phosphatase in BAT thermogenesis (**line 102-104**). Cold stimulation induces β_3 -adrenergic signaling, especially in thermogenic adipocytes (brown and beige adipocytes) that express β_3 -adrenergic receptor (Cypess, A. M. et al. *Cell Metab* 2015, Inagaki, T. et al. *Nat Rev Mol Cell Biol* 2016). In addition, when exposed to cold, brown adipocytes are activated within a short period of time via the sympathetic nervous system to rapidly adapt to changes in ambient temperature (Ouellet, V. et al. *J Clin Invest* 2012, Betz, M. J. et al. *Nat Rev Endocrinol* 2018). In this paper, we revealed that Ssu72 expression was enriched in BAT at RT and increased in response to acute cold. We also found that Ssu72 expression was also increased in iWAT of WT mice after 12 hours of cold exposure (**Supplementary Fig. 2d**). However, Ssu72 aKO mice exhibited a rapid loss of body temperature and fatal hypothermia within 6 hours of exposing to severe cold (**Fig. 2a** and revised **Supplementary Fig. 10c**). Therefore, cold susceptibility of Ssu72 aKO mice during acute cold exposure was likely affected by brown adipocyte dysfunction.

It should be noted that interfering with any important cellular process usually has an effect on mitochondria and consequently has a physiological phenotype. Interfering with global translation in a specific tissue is for sure going to disrupt mitochondrial functions and obstruct the normal function of the tissue. The only solid and novel conclusion of the manuscript is that Ssu72 is a phosphatase of eIF2 α .

Response: We have carefully considered the point raised by the reviewer. Regulation of eIF2 α phosphorylation by Ssu72 in BAT appears to be important for maintaining body temperature through translation of key effectors of thermogenesis. To address the reviewer's question, we have investigated effects of Ssu72 on mitochondrial function and thermogenesis in a non-adipose tissue such as liver, where mitochondrial function is important for metabolic health of the tissue. We used hepatocyte-specific Ssu72 knockout mouse model (Albumin-Cre; Ssu72^{flox/flox}) for cold challenge experiment. After Ssu72^{flox/flox} (Ssu72^{f/f}) and Albumin-Cre; Ssu72^{flox/flox} (Alb-Cre; Ssu72^{f/f}) mice were exposed to acute severe cold (4°C) for 6 hours, we found that rectal temperatures of Alb-Cre; Ssu72^{f/f} mice were similar to those of Ssu72^{f/f} mice during cold exposure (shown below, **Fig. R5a**). This result indicates that cold tolerance is not altered by hepatocyte-specific Ssu72 deletion. Considering the points raised by the reviewer, we additionally determined changes in phosphorylation status of eIF2 α and RNAPII CTD by Ssu72 deletion in the liver. Interestingly, as shown in **Fig. R5b**, there was no significant difference in the eIF2 α phosphorylation level by Ssu72 depletion in liver. In addition, we have

revealed that the effect Ssu72 on eIF2 α dephosphorylation, especially in adipocytes, might be greater than those of other eIF2 α phosphatases (revised **Supplementary Fig. 8b**). These results imply that the regulation of eIF2 α by Ssu72 phosphatase is more likely to be regulated in a tissue-specific manner rather than a global regulation. When comparing phosphorylation levels of both Ser5 and Ser7 of RNAPII CTD in hepatocytes, P-Ser5 showed a slight increase, but P-Ser7 did not show a significant difference (shown below, **Fig. R5b**). Thus, hepatic transcriptional regulation is not significantly affected by Ssu72.

Not shown in the manuscript

Fig. R5 (Not shown in the manuscript): **a**, Rectal core body temperatures of 7-8 weeks old Ssu72^{f/f} and Albumin-Cre; Ssu72^{f/f} (Alb-Cre; Ssu72^{f/f}) mice under cold conditions (4°C) (n = 4-5 mice per group). Data are presented as mean \pm SEM. Statistical comparisons were made using multiple t-tests. **b**, Western blots of lysates from liver of 8-week-old Ssu72^{f/f} and Alb-Cre; Ssu72^{f/f} littermates exposed to acute cold (4°C) for 6 hours.

We further investigated expression levels of mitochondrial OXPHOS complex subunits in liver of Ssu72^{f/f} and Alb-Cre; Ssu72^{f/f} mice. We found that there was no significant difference in

Not shown in the manuscript

Fig. R6 (Not shown in the manuscript): Western blots of OXPHOS complex subunits in liver of 8-week-old Ssu72^{f/f} and Alb-Cre; Ssu72^{f/f} littermates exposed to acute cold (4°C) for 6 hours.

protein expression of OXPHOS complex between the genotypes, as shown in **Fig. R6**. Together, these results demonstrate that cold-responsive Ssu72 is involved in the translation of key thermogenic effectors, particularly in BAT, thus affecting mitochondrial function and thermogenesis.

In addition to being the phosphatase of eIF2 α , Ssu72 is also the phosphatase of RNAPII and probably many more, the authors haven't taken the effects of these other substrates and the corresponding pathways into consideration in their explanation of the knockout phenotypes.

Response: We agree with the reviewer that we should have considered effects of other Ssu72 substrates on BAT function. To address this point, we have investigated changes in phosphorylation status of other Ssu72 substrates such as RNAPII in BAT of Ss72 WT and aKO mice. While only P-Ser7 was slightly increased in Ssu72 aKO mice under basal conditions, RNAPII CTD phosphorylation (P-Ser5, P-Ser7) under cold conditions was not significantly different between WT and aKO mice (shown below, **Fig. R7a, b**), indicating that RNAPII transcriptional regulation by Ssu72 is not significantly affected in BAT. Additionally, protein expression of HNF4a, previously identified as a Ssu72 substrate (Kim, H. et al. *Cell Death Differ* 2022), was too low in BAT to compare the expression levels between genotypes (shown below, **Fig. R7c**).

Fig. R7 (Not shown in the manuscript): **a, b** Western blots in BAT of Ssu72 WT and aKO mice housed at RT (**a**) or exposed to acute cold (4°C) for 4 hours (**b**) (n = 3 mice per group). **c**, Western blots of HNF4 α in BAT from WT and aKO mice (n = 3) and liver from WT mice housed at RT. The liver of WT mice was used as positive control for HNF4 α expression.

We hypothesized that this result might be due to different localization of Ssu72 in brown adipocytes. In brown adipocytes, Ssu72 is predominantly localized in the cytoplasm (shown below, **Fig. R8a**). Different localization of Ssu72 in brown adipocytes was further confirmed by subcellular fractionation of BAT and liver of WT mice. In the liver, Ssu72 was detected in the fractionated extracts of both cytosol (C) and nucleus (N) by immunoblotting, and Ssu72

was more abundant in the cytosol (shown below, **Fig. R8b**). Interestingly, while some Ssu72 were detected in the nuclear fraction of liver lysates, most Ssu72 were detected in the cytosolic fraction in BAT (shown below, **Fig. R8b**). Therefore, in brown adipocytes, Ssu72 appears to play an important role in the cytosol. Collectively, novel findings of this study are that Ssu72 expression is increased in BAT in response to acute cold and that Ssu72 is involved in the translation of key thermogenic effectors such as OXPHOS complex by eIF2 α dephosphorylation in BAT, thereby maintaining mitochondrial function and thermogenesis during cold exposure.

Not shown in the manuscript

Fig. R8 (Not shown in the manuscript): a, Immunohistochemical analysis with α -Ssu72 in BATs of 12-week-old Ssu72 WT and aKO mice housed at RT. (BAT of Ssu72 aKO mice were used as negative control) **b**, Western blot of lysate from subcellular compartments (cytosol (C) and nucleus (N)) of BAT and liver of WT mice housed at RT.

There are also some additional concerns:

1. *Fig. 1b, Ssu72 is not expressed only in BAT. Hearts probably have higher expression levels. How specific this response to temperature in BAT is hasn't been looked at.*

Response: As mentioned above, cold exposure specifically induces β_3 -adrenergic signaling pathway in thermogenic adipose tissue. In addition, we found that expression levels of

mitochondrial OXPHOS complex subunits in other metabolic tissues, including heart, skeletal muscle, and liver, were not altered by cold exposure for 2 days (shown below, **Fig. R9**).

Not shown in the manuscript

Fig. R9 (Not shown in the manuscript): Western blots of mitochondrial OXPHOS complexes (CI-V) in heart, skeletal muscle, and liver of WT mice exposed to RT or Cold for 2 days.

BAT thermogenesis is essential for mammals to rapidly maintain body temperature during acute cold exposure. We showed that *Ssu72* expression is increased in BAT by acute cold exposure (revised **Fig. 1c,d**). To clarify the specific response of *Ssu72* to cold in BAT, we assessed changes in *Ssu72* expression in response to cold in other metabolic tissues. We found that expression levels of *Ssu72* in response to acute cold for 5 hours in other metabolic tissues (heart, skeletal muscle, and liver) were similar to those at RT (shown below, **Fig. R10**). These results indicate that *Ssu72* expression is upregulated upon cold exposure, especially in BAT.

Not shown in the manuscript

Fig. R10 (Not shown in the manuscript): WT mice housed at RT or exposed to cold (4°C) for 5 hours (n = 4 mice per group) **a**, Relative mRNA expression levels of *Ucp1* in BAT from WT mice housed at RT or exposed to cold. mean \pm SEM. ***p < 0.001 (Student's t-test). **b-d**, Western blots of *Ssu72* in heart (**b**), skeletal muscle (**c**), and liver (**d**) of WT mice housed at RT or exposed cold.

2. Fig. 1d, GAPDH is a metabolic protein. Is it a good control in this case? From the figures shown, GAPDH levels at RT and cold are clearly not the same.

Response: We agree with the reviewer that GAPDH is a metabolic protein. Indeed, GAPDH has been used as a Western blot loading control in BAT during acute cold exposure (Liu, D. et al. *J Clin Invest* 2016, Chitraju, C. et al. *Cell Rep* 2020) and chronic cold exposure (Shin, H. et al. *Cell Metab* 2017, Yau, W. W. et al. *iScience* 2021). To clarify the suitability of GAPDH as a loading control in BAT during cold exposure, we have assessed expression levels of other loading controls (β -actin and α -tubulin) in the same sample for comparison. β -actin and α -tubulin have also been used as loading controls in BAT during acute cold exposure (Albert, V. et al. *EMBO Mol Med* 2016, Han, J. S. et al. *Nat Commun* 2022). Considering the difference in expression level of each loading control under RT and cold conditions, we have further quantified relative Ssu72 protein expression levels normalized to each control. We found significant differences in Ssu72 expression between RT and cold conditions (Shown below, **Fig. R11**).

Relative to Fig. 1d

Fig. R11 (Relative to Fig. 1d): Western blots of lysates from BAT of C57BL/6J mice housed at RT (23°C) or exposed to acute cold (4°C) for 4 hours (left) (n = 3 mice per group), with quantification of Ssu72 levels (right). Graph shows quantification of Ssu72 protein levels normalized to GAPDH, β -actin, or α -tubulin protein level using ImageJ. Data are presented as mean \pm SEM. **p < 0.01, ***p < 0.001. Statistical comparisons were made using Student's t-test.

3. Fig. 2d, again, why use the heat shock protein as a control if the authors were studying cold response?

Response: We thank the reviewer for raising this important point. Some heat shock proteins are constitutively expressed and can be used as loading controls, but it is important to consider that certain conditions can alter protein expression. It is noteworthy that cold stimulation specifically induces β_3 -adrenergic signaling in adipocytes, and HSP90 protein expression is not altered by β_3 -adrenergic signaling in adipocytes during acute cold exposure (Gerhart-Hines, Z. et al. *Nature* 2013, Paulo, E. et al. *Sci Rep* 2018) or chronic cold exposure (Dijk, W. et al. *Elife* 2015, Kutuyavin, V. I. et al. *Proc Natl Acad Sci U S A* 2019). To clarify the suitability of HSP90 as a loading control in BAT, we have newly added expression of β -actin. We found a similar expression pattern between HSP90 and β -actin during acute cold exposure (**Fig. 2d**).

4. Fig. 4d, are the two lanes two animals? Clearly HSP90 levels are very different in the two WT samples.

Response: We thank the reviewer for this comment. The number of mice per group is two and each lane represents each mouse. We have also described information about the number of mice in each figure legend. To show clearer data, we have repeated the experiment considering the amount of loading control (HSP90) and replaced it with new data (**Fig. 4d**).

5. Fig. 4f, the myc control also has some clear HA signal. Myc-Ssu72 IP controls are not equal either.

Response: We apologize for this confusion. The reason why some HA signal appeared in the myc control under IP condition was probably because washing of immunoprecipitated proteins during IP assay was not sufficient. Therefore, we have repeated the Co-IP assay and replaced this figure with new data (**Fig. 4f**).

6. Fig. 6b, the quantification does not match the blots, especially for AMPK α .

Response: We have newly added results of other loading controls (β -actin and α -tubulin) and further quantified relative expression levels of thermogenic proteins (PGC-1 α , UCP1, AMPK α , PKA) normalized to the expression level of α -tubulin (**Fig. 6b**).

7. Fig. 6c, and d, the Ssu72 blots do not show a three-fold increase of aKO;cTg compared to WT as shown in quantification graph.

Response: We apologize for the confusion. Fig. 6c is the result of protein expression in BAT and Fig. 6d in the previous manuscript (shown in revised **Supplementary Fig. 10a**) is mRNA expression result, not the quantification graph of protein levels in Fig. 6c. Thus, although expression level of Ssu72 mRNA in aKO;cTg mice was increased compared to WT (revised **Supplementary Fig. 10a**), there was no significant difference in protein expression level of Ssu72 between WT and aKO;cTg mice in the same mouse groups (**Fig. 6c**). The reason why we presented mRNA expression levels of thermogenic genes in aKO;cTg mice was to show that transcription of thermogenic factors in aKO;cTg mice was not significantly different from that in WT mice under RT conditions (revised **Supplementary Fig. 10a**).

8. Fig. 6, 7 and extended data Fig. 9, most of these results only have RT data. No cold exposure data were shown. If there was a three-fold increase of Ssu72 in aKO;cTg as claimed, which is probably more relevant to the increase of Ssu72 levels in BAT after cold exposure, shouldn't the levels of the examined proteins be higher in aKO;cTg compared to WT in extended data Fig 9a?

Response: Ssu72 mRNA expression level was increased about 3-fold in aKO;cTg mice compared to WT mice (revised **Supplementary Fig. 10a**). However, this increase did not affect

Ssu72 protein expression in aKO;cTg mice, which showed a similar expression level to WT mice (**Fig. 6c**). The reason why we mainly showed RT data is that expression levels of some thermogenic proteins were already reduced in BAT of Ssu72 aKO mice under RT condition (**Fig. 6b**), and expression levels of OXPHOS complexes, which are essential enzymes for thermogenesis, were also decreased in BAT of Ssu72 aKO mice at RT (revised **Fig. 7c, e**). These results suggest that Ssu72 aKO mice are insufficient to induce thermogenesis in response to severe cold exposure. In aKO;cTg mice, COX4 proteins were not completely restored to the expression levels in WT mice (revised **Fig. 7f**). However, after 4 hours of cold exposure (4°C), the sluggish and shivering behavior of Ssu72 aKO mice was restored by ectopic Ssu72 expression in aKO;cTg mice (**Supplementary Video 1**). In addition, we assessed UCP1 expression in BAT during cold exposure and found that reduced protein expression of UCP1 in aKO mice was also restored in aKO;cTg mice under cold condition (revised **Fig. 6e**).

Point-to-Point Responses to the Reviewers' Comments (#NCOMMS-22-32682-A)

Reviewer #2 (Remarks to the Author):

The authors have addressed most of my concerns. The new results have shown that cold exposure has no effect on Ssu72 expression in all other tissues but BAT and iWAT and that knockdown of Ssu72 has no effect in all the substrates in any examined tissues except in BAT, and no effect in any other substrates in BAT except eIF2 α . This kind of specificity is very interesting. The authors did attempt to find an explanation. However, Fig. R2B does not show the conclusion as stated in the response because the nuclear controls are not even. Even if there were minor subcellular Ssu72 localization differences between the two tissues, it still does not explain the big function differences of the major cytosolic pools in the two tissues. Personally, I don't think this question has to be addressed for the paper if the other conclusions are solid. I would like to see, however, how UCP1 levels are changed in all the other tissues, since UCP1 levels have been the major indicating factor throughout the manuscript. In addition, the authors have done many experiments to address the concerns, but left the results out of the manuscript. I think it would be nice to present a more complete set of data and let the readers draw their own conclusion, instead of having the selected set and trying to convince readers of a conclusion that was actually conditional. This, however, is for the editors to decide.

Response: We thank the reviewer for this comment. Since liver is a tissue that responds to various stress signals, many studies have investigated ER stress/eIF2 α signaling in liver. However, we found that Ssu72 deletion in hepatocytes did not significantly affect eIF2 α phosphorylation. This might be due to different regulatory mechanisms in eIF2 α signaling by other well-known eIF2 α phosphatases (GADD34, CReP, PP1). Interestingly, these eIF2 α phosphatases were highly expressed in various cell-types within tissues, but relatively low in adipocytes. On the other hand, Ssu72 is highly abundant in adipocytes (revised **Supplementary Fig. 8b**). These results suggest that ER stress/eIF2 α signaling pathway is regulated in non-adipose tissues because other phosphatases have a relatively high effect on eIF2 α dephosphorylation, but in BAT, Ssu72 is essential for regulating eIF2 α phosphorylation. In addition, we also found that HFN4 α , identified as a target of Ssu72 in hepatocytes (Kim, H. et al. *Cell Death Differ* 2022), was rarely expressed in BAT as shown in **Fig. R7c**, suggesting that specificity of Ssu72 function may differ between tissues depending on the presence of Ssu72's target proteins in each tissue. Following the reviewer's suggestion, we have evaluated UCP1 expression in other tissues. We found that the UCP1 protein expression in both muscle and liver was relatively lower than in BAT (shown below, **Fig. R12a**). In addition, when assessing for changes in UCP1 expression by cold exposure in other metabolic tissues such as heart, skeletal muscle, and liver, it was found that UCP1 protein expression was not increased in response to acute cold in these tissues (shown below, **Fig. R12b-d**). This result suggests that UCP1, a key thermogenic factor, is more specifically expressed and acts on thermogenesis in BAT than in other metabolic tissues during acute cold exposure.

Not shown in the manuscript

Fig. R12 (Not shown in the manuscript): C57BL/6J (WT) mice housed at RT (23°C) or exposed to cold (4°C) for 5 hours. **a**, Western blots of UCP1 and α -tubulin (loading control) in skeletal muscle, liver, and BAT of WT mice housed at RT. **b-d**, Western blots of UCP1 and β -actin (loading control) in heart (**b**), skeletal muscle (**c**), and liver (**d**) of WT mice housed at RT or exposed cold (n = 4 mice per group).